

# Entanglement flow in the Kane-Fisher quantum impurity problem

Chunyu Tan[1], Yuxiao Hang[1*], Stephan Haas[1] and Hubert Saleur[1,2]

**1** Department of Physics and Astronomy, University of Southern California,
Los Angeles, CA 90089-0484, USA
**2** Institut de Physique Théorique, Université Paris Saclay,
CEA, CNRS, F-91191 Gif-sur-Yvette, France

★ yhang@usc.edu

## Abstract

The problem of a local impurity in a Luttinger liquid, just like the anisotropic Kondo problem (of which it is technically a cousin), describes many different physical systems. As shown [1] by Kane and Fisher, the presence of interactions profoundly modifies the physics familiar from Fermi liquid theory, and leads to non-intuitive features, best described in the Renormalization Group language (RG), such as flows towards healed or split fixed points. While this problem has been studied for many years using more traditional condensed matter approaches, it remains somewhat mysterious from the point of view of entanglement [2], both for technical and conceptual reasons. We propose and explore in this paper a new way to think of this important aspect. We use the realization of the Kane Fisher universality class provided by an XXZ spin chain with a modified bond strength between two sites, and explore the difference of (Von Neumann) entanglement entropies of a region of length $\ell$ with the rest of the system - to which it is connected with a modified bond - in the cases when $\ell$ is even and odd. Surprisingly, we find out that this difference $\delta S \equiv S^e - S^o$ remains of $O(1)$ in the thermodynamic limit, and gives rise now, depending on the sign of the interactions, to "resonance" curves, interpolating between $-\ln 2$ and $0$, and depending on the product $\ell T_B$, where $1/T_B$ is a characteristic length scale akin to the Kondo length in Kondo problems [3]. $\delta S$ can be interpreted as a measure of the hybridization of the left-over spin in odd length subsystems with the "bath" constituted by the rest of the chain. The problem is studied both numerically using DMRG and analytically near the healed and split fixed points. Interestingly - and in contrast with what happens in other impurity problems [4] - $\delta S$ can, at least to lowest order, be tackled by conformal perturbation theory.

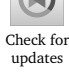

# 1  Introduction

## 1.1  Generalities

The effect of localized impurities on interacting systems of condensed matter physics is one of the most studied incarnations of true many body physics, with a wealth of related theoretical developments (see e.g. [5–8]) as well as experimental applications (see e.g. [6, 9, 10] for general reviews, or [11, 12] for more recent work). Maybe no such incarnation is more spectacular than the Kane Fisher problem [1] - which can be considered as the effect of a localized impurity in a 1D Luttinger liquid, or equivalently, via the magic of the Jordan-Wigner transformation, the effect of a modified bond in an XXZ spin chain. Thanks to more reformulations, this problem finds experimental applications in the physics of one-channel conductors in an Ohmic environment [13], in the physics of tunneling between edges in the fractional quantum Hall effect [14–16], in the physics of quantum Brownian motion [17], in the physics of small Josephson junctions [18], and more. It has also recently given rise to quantum simulations as in [19, 20].

The main physics of the Kane-Fisher problem can be expressed in a nutshell through the renormalization group language: in the presence of repulsive (resp. attractive) interactions between $1d$ spinless electrons, an arbitrarily weak impurity flows to a large barrier (resp. becomes invisible) at low-energy. In what follows, we will use the equivalent language of antiferromagnetic spin chains of XXZ type, where repulsive (resp. attractive) corresponds to the $J_z$ coupling being positive (resp. negative). In this context, it is customary [21] to refer to the two possible end points of the RG flow as split (resp. healed). Note of course the sharp difference with the non-interacting case, where an impurity behaves as the same barrier at all scales, leading to energy independent properties.

Of course the RG flow is just a tool to encapsulate the behavior of the system at low-energy. Low-energy may mean, depending on the context, low temperatures (e.g. when considering the conductance of the $1d$ electron gas), large distances (e.g. in considering Friedel oscillations [22,23]), long times (e.g. in considering quenches [24,25]), or low frequency (e.g. in considering shot noise features [26,27]). Note also that healing at low energy in the Kane Fisher problem is similar qualitatively to what happens in the Kondo model, where the impurity is absorbed by the electron bath at low-energy. The two problems are in fact also very close technically [28–30].

Among all the probes at our disposal to study this physics, entanglement - while one of the most fundamental as it directly probes the nature of the wave functions themselves - is one of the most difficult to address (let alone measure experimentally). While it would be natural, in quantum impurity problems, to consider measures of the entanglement of the impurity with the rest of the system - for instance, in the XXZ language, entanglement of the two spins across the impurity bond with the rest of the chain, or, in the Kondo case, entanglement of the impurity spin with the electron bath - this quantity turns out not to present any interesting crossover, ultimately because it (via the reduced density matrix) involves only a finite number of particles (or spins) [2]. To get a non-trivial crossover, one needs to explore the whole extent of the screening cloud [31], i.e. have another length scale in the system. This is technically difficult. A possibility is to consider the entanglement of a region of length $\ell$ centered on the impurity with the rest of the system [32]: in this case, the RG flows affects the behavior of the $O(1)$ terms, which are technically difficult to extract from data. Another possibility is to consider the entanglement between two regions separated by the impurity. This was considered in [33] (and in [34] in the random case, where the physics is partly, but not entirely different). In this case, the RG flow affects the (slope of the) entanglement linear dependency with $\ln \ell$.

We propose in this paper another way to characterize the RG flow using entanglement for the Kane-Fisher problem. The idea mimics, in a certain sense, what happens in the Kondo model: isolated regions of length $\ell$ contain, when $\ell$ is odd, a "left-over" spin (since their ground-state is twice degenerate), which should get hybridized with the "bath" constituted by the other spins in an infinite chain when the region is coupled to it by a modified bond. No such hybridization should take place when $\ell$ is even. This suggests comparing the entanglement of subsystems of even and odd length, and, specifically, studying how their difference $\delta S$ behaves along the RG flow. As we shall see below, $\delta S$ indeed exhibits the expected behavior, interpolating between $-\ln 2$ and $0$ (or the other way around, depending on the interactions) as energy is lowered, and providing an interesting new way to probe the underlying physics.

We emphasize that, apart from this physical motivation, the study of quantities such as $\delta S$ – which is, technically, a term of $O(1)$ in the entanglement – is of ever increasing interest. These terms encode indeed features of topological phases in higher dimensions [35], or of topological defects in one dimension [36]. Issues about the role of the position of the entanglement cut [37], or the boundary conditions induced in the effective field theory by this cut [38] on such terms of $O(1)$ have been raised [39], as well as questions about the role of zero modes and finite-size effects [40,41], or of localized excitations [42]. We will see in this paper that terms of $O(1)$ can arise due to parity effects, and, moreover, can be controlled analytically using conformal perturbation theory.

## 1.2 Set-up in this paper

The simplest way to observe parity effects in the Kane-Fisher problem is to consider an impurity at a distance $\ell$ from the boundary of the system (a similar effect without boundary would require two impurities, as discussed in the conclusion). In an earlier work [43] we studied such parity effects for the XX chain where, however, a modified bond behaves as a conformal

defect, and there is no RG flow. In this paper, we extend the analysis of [43] to the interacting case, where the physics, as discussed above, is profoundly different

To be more specific, the system we consider here is an XXZ spin chain of length $L$ with free boundary conditions at either end, and a modified coupling at position $\ell$ (see below for a fully accurate description of the model). This is equivalent, thanks to the Jordan-Wigner transformation, to a Luttinger liquid with a modified tunneling amplitude on one bond (and, in some variants, a modified Coulomb interaction, see below).[1] Without interactions - i.e., in the XX chain - the modified coupling translates into an exactly marginal perturbation from an RG point of view. The entanglement of the region $A$ starting at the boundary and ending at the modified bond was found in in this case [43] to possess, on top of the expected $\ln \ell$ term with a factor proportional to the effective, coupling dependent central charge, a term of $O(1)$ that differs between the even and the odd cases. The corresponding difference $\delta S$ was found to be a universal function with interesting properties, interpolating between $-\ln 2$ and 0. In the case with interactions, the modified coupling induces, as discussed above, an RG flow, with two possible fixed points, the fully split chain and the healed chain. Depending on the sign of $J_z$, one of these is stable and the other one unstable. Under a relevant perturbation of the stable fixed point by a modified coupling at position $\ell$, the entanglement now has a $\ln \ell$ term whose slope depends on an effective central charge, and can be expressed as a function of $\ell T_B$ where $T_B$ is a characteristic energy scale akin to the Kondo temperature in Kondo-like impurity problems. It is not this slope we are interested in here, but rather the terms of $O(1)$, which turn out to differ between the even and odd cases, while the corresponding difference $\delta S$ is now a universal quantity depending, like the effective central charge, on the product $\ell T_B$. Just as in the non-interacting case $J_z = 0$, $\delta S$ interpolates between $-\ln 2$ and 0.

As commented in the conclusions, similar parity effects would be observed in the periodic case: we emphasize they have conceptually nothing to do with the presence of a boundary. It is interesting to reflect a bit more on their physical meaning. The best way to start is to think of Kondo physics from the point of view of entanglement. As mentioned earlier, while it is tempting to think that, as the Kondo impurity gets screened by conduction electrons at low energy, it becomes more entangled with them, this clearly cannot be: the entanglement of the Kondo impurity as measured by the Von Neumann entropy of the corresponding spin with the rest of the system (the "single site impurity entanglement entropy" [2]) is (in the absence of a magnetic field) fixed at $\ln 2$ irrespective of the Kondo coupling. To be able to define non-trivial quantities (that can be used later on to provide signatures of the screening cloud [44]) one needs to introduce another (length) scale. In the Kondo literature, it is common to consider for this an interval (in the s-wave language) of the system extending a distance $r$ from the impurity [2,45], or, in related purely one-dimensional problems, an interval of length $L$ centered on the impurity [4]. The physics at play can then be understood in terms of valence bonds originating from the impurity and reaching out to the rest of the system. Roughly, the "impurity part" [2] of the entanglement reaches $\ln 2$ when $r \lesssim \xi_K$ (with $\xi_K$ the Kondo length) that is, when $r$ is smaller than the typical length of the valence bond originating from the impurity spin.

The physics we have in our case is somewhat similar and could be intuitively interpreted in a valence bond picture [46]. Think for instance of the Hamiltonian (1) below. As illustrated in Fig. 1(a), in the limit of very small $\lambda$ (corresponding to a very small impurity bond) and in the simplified valence bond picture, valence bonds "prefer" not to stretch over the weak link: only one is forced to do so in the odd length case, contributing $-\ln 2$ to the difference of entanglements between even and odd. On the contrary, if $\lambda \sim 1$, there are a lot of such valence bonds, and even though the parity of their numbers is odd in the odd case and even in the even case, the difference averages to a small term that decays with $L$. Fig. 1(b) shows the constant terms in even and odd cases when $J_z = 0$ which confirms our qualitative arguments (it is difficult

---

[1]In what follows, we will freely use interchangeably the spin or electron language.

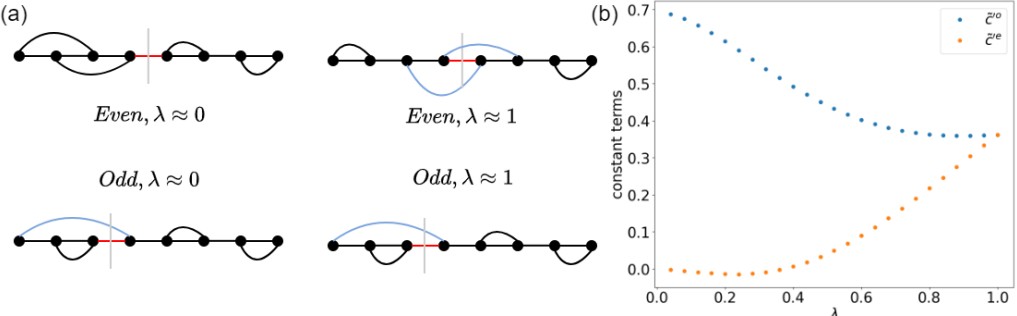

Figure 1: (a). Valence bond formation for $\lambda \approx 0$ ($\lambda \approx 1$) for both even and odd cases. The bonds separating the regions of the bipartition are illustrated by grey lines, and inter- (intra-) subsystems valence bonds are depicted by black (dark blue) curved lines. (b). The constant ($\ell$ independent) $O(1)$ terms for even and odd subsystems when $J_z = 0$.

to define unambiguously the $O(1)$ terms when $J_z \neq 0$, see below). In the intermediate region, the result depends on the average length of the valence bonds, which in this interpretation becomes $1/T_B$, the equivalent of $\xi_K$.

Interestingly, while the underlying physics is the one of the Kane-Fisher [1, 47] problem, from the point of view of entanglement we have results akin to the Kondo problem, with entanglement curves always interpolating between $-\ln 2$ and 0, and a "resonance" as can be seen e.g. in Fig. 4. In a certain sense, we are thus re-interpreting the Kane-Fisher problem as screening of the delocalized spin $1/2$ in our subsystem of odd length.

## 1.3 Organization

The paper is organized as follows. In section 2, we discuss the vicinity of the split fixed point and in section 3 the vicinity of the healed fixed point. In both cases, we provide "ab-initio" numerical results together with comparison with (conformal) perturbative calculations, especially of the entanglement. Most detailed calculations are done in subsections 3.3,3.5 where some subtle aspects - including the renormalization between bare and renormalized couplings - are investigated in detail. We find in particular that, even though the entanglement cut and the location of the perturbation essentially coincide, no non-universal effects seem to be encountered. This is in contrast with the current expectation for entanglement across topological defects [48] when the cut is at the same location as the defect. In section 4 we discuss several interesting symmetry properties obeyed by $\delta S$. Section 5 contains a few conclusions. together with a preliminary discussion of what happens in the case without boundary. A short appendix provides technical details about the DMRG calculations.

Finally, to help the reader who may not be familiar with some of the vocabulary we use we recall below the definition of a few terms [21]:

- Split and healed: refers to a spin chain with one (or several) bond being associated to a coupling of modified strength. When this coupling vanishes, the chain is split: in the RG language, this corresponds to a fixed point referred to as "split fixed point". When the modified coupling is identical with the others, the chain is homogeneous: in the RG language, this happens to correspond to another fixed point which we call "healed fixed point".

- UV and IR: we refer to the high-energy or short distance or short time regions of the parameter space as the UV region, and the low-energy or large distances or long times as the IR region.

We also emphasize that this paper deals only with the case where the defect is *not conformal*, and thus $J_z \neq 0$. The non-interacting case $J_z = 0$ corresponds to a conformal defect instead, where the is no RG flow, and the bulk behavior of the entanglement involves an interaction dependent central charge [49]. Terms of $O(1)$ in this case have been studied in [43].

## 2 Physics around the split fixed-point

We start by considering physics around the split fixed-point, that is, when the chain is almost cut in half at the impurity bond. After a qualitative general discussion, we report numerical results for both the relevant and irrelevant cases. We then carry out a conformal perturbation calculation of the shifts in energy and entanglement due to the impurity. We show in particular that, while odd and even lengths $\ell$ give rise to different functional forms for the energy-shift, the entanglement has a similar analytic structure in both cases, so that $\delta S$ can be properly defined and expected to be universal.

### 2.1 Generalities

We consider first the Hamiltonian

$$
\begin{aligned}
H^A = \sum_{j=1}^{\ell} \left( \sigma_j^x \sigma_{j+1}^x + \sigma_j^y \sigma_{j+1}^y + J_z \sigma_j^z \sigma_{j+1}^z \right) + \lambda \left( \sigma_\ell^x \sigma_{\ell+1}^x + \sigma_\ell^y \sigma_{\ell+1}^y + J_z \sigma_\ell^z \sigma_{\ell+1}^z \right) \\
+ \sum_{j=\ell+1}^{L} \left( \sigma_j^x \sigma_{j+1}^x + \sigma_j^y \sigma_{j+1}^y + J_z \sigma_j^z \sigma_{j+1}^z \right),
\end{aligned}
\tag{1}
$$

where the $\sigma$'s are Pauli matrices. This Hamiltonian is made of two XXZ chains of length $\ell$ and $L - \ell$ respectively, with a modified interaction between them. In (1) we take the same form of the modified interaction as in the bulk, but with a different amplitude $\lambda$. Later we shall also consider a modified interaction different from the one in the bulk, and of the type $\sigma^x \sigma^x + \sigma^y \sigma^y$, see below (eq. (8)). We shall only be interested in the regime $-1 < J_z \leq 1$ (where the chain is gapless), and in properties in the scaling limit, where $\ell, L \to \infty$. Details of the results in this limit will depend on the ratio $L/\ell \equiv Z$, but will be qualitatively independent of $Z$. We will in what follows mostly (but not only) consider $Z = 2$.

In the regime we are interested in, XXZ Hamiltonians can be mapped onto a free boson theory at low-energy (this is discussed more below). The various local operators can then be reformulated into exponentials of the free boson and its derivatives. The conformal dimensions depend on a single parameter which is sometimes taken to be the "Luttinger liquid coupling" [50], the "boson radius" (see e.g. I. Affleck's lectures in [51]) or the "coupling constant" [52] - this is the wording (and associated notation) we shall use below. Setting

$$
g = 2 - \frac{2}{\pi} \arccos(J_z),
\tag{2}
$$

we see that $g > 1$ for $J_z > 0$ and $g < 1$ for $J_z < 0$. Near what we call the *split fixed point* $\lambda = 0$, a small nearest-neighbor interaction as in (1) can be interpreted as the coupling of a local bulk operator of dimension $(length)^{-g}$. This means that

near the split fixed point $\lambda$ is relevant if $J_z < 0$,

near the split fixed point $\lambda$ is irrelevant if $J_z > 0$. (3)

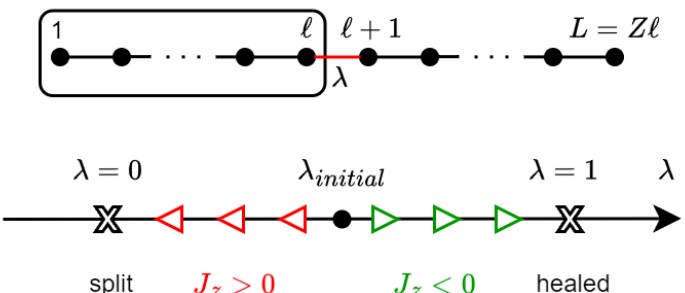

Figure 2: The different types of flows near the split fixed-point ($\lambda$ small). In one case the system appears totally split at low energy, while in the other it appears homogeneous ("healed").

The RG flows are thus as in Fig. 2. Writing the perturbation as $\lambda\mathcal{O}$, this product must have dimension (length)$^{-1}$ and thus, if $\mathcal{O}$ has dimension (length)$^{-g}$, we find

$$\dim\left[\lambda\right] = (\text{length})^{g-1}. \tag{4}$$

Hence we can construct a quantity of dimension (length)$^{-1}$ (a temperature) by considering[2]

$$T_B \equiv \lambda^{1/(1-g)}. \tag{5}$$

In the relevant case, it is well known [3] that the chain at large distances appears *healed* - meaning, its physics is the same as the one of a homogeneous chain. This can be clearly seen if we consider the entanglement of the region of size $\ell$ to the left of the cut with the rest of the system: the leading ("bulk") behavior of the entanglement should interpolate from $S = 0$ to $S \approx \frac{1}{6}\ln\ell$ as $\ell$ is increased at fixed $\lambda$ (for earlier studies of this problem, see [53]). This is illustrated in Fig. 3 where we have plotted the derivative of the entanglement entropy for the system in the particular case $Z = 2$ ($L = 2\ell$) when $J_z = $ -0.5. Recall that, from finite size scaling results, the entanglement in the healed case always has the leading behavior $\frac{c}{6}\ln L$ with $c = 1$ here. This is seen in Fig. 3 as the the two curves approach $6dS/d\ln L = 1$ at large distances (that is, in the IR). Note that the results look quite different for odd and even lengths $\ell$ (represented by $S^e$ and $S^o$ respectively). It is this difference we shall be interested in in what follows.

We note that IR physics which we observe here using finite length effects in equilibrium could also be studied with a quench set-up by going at large times. See for instance [25,33] - in these references however, parity effects are not investigated.

Like in [43] we expect to have, for an infinite system (more complicated formulas involving $\ell/L$ are required for a finite system, see below)

$$\frac{dS^e_{A,\text{imp+bdr}}}{d\ln\ell} = F(\ell T_B) + f^e(\ell T_B),$$
$$\frac{dS^o_{A,\text{imp+bdr}}}{d\ln\ell} = F(\ell T_B) + f^o(\ell T_B), \tag{6}$$

where $F, f^e, f^o$ are non-trivial, universal functions. Note that in [43], the forms (6) were encountered when considering a non-interacting system with a dot-like impurity - that is, two

---

[2]Note that definitions of $T_B$ differing by numerical factors may appear in the literature.

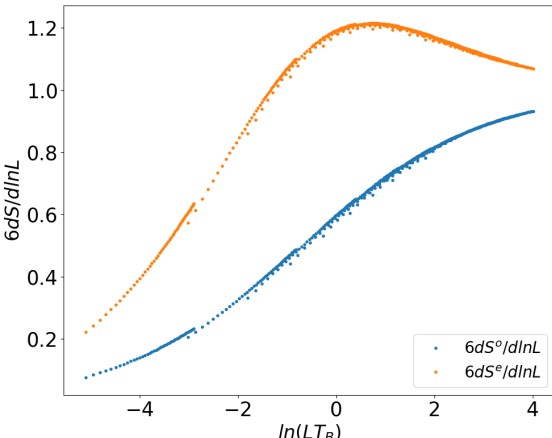

Figure 3: Crossover in the (slope of) the bulk entanglement entropy for Hamiltonian 1 illustrating healing at low-energy (meaning, the slope goes to $\frac{1}{6}$) when $J_z = -0.5$. $T_B$ is defined in (5).

successive bonds modified. This case gave rise to a non-trivial RG flow, just like the case of a single modified bond in the presence of interactions that we study in this paper.

In (6), $F$ encodes a sort of effective central charge, while $f^{e,o}$ are "terms of $O(1)$". This name might sound inadequate from (6), but as discussed in detail in [54], only derivatives of the entanglement obey proper scaling. After taking derivatives, the (parity independent) "bulk" term $\frac{c}{6} \ln \ell$ ceases to dominate the expression for the entanglement, due to the fact that $\frac{d}{d \ln \ell} \ln \ell = 1$. Like in our earlier work [43] we shall focus on the difference $f^e - f^o$, which originates from terms in $S^{e,o}$ whose difference does not decay as $\ell \to \infty$ (as illustrated on figure 3). In the following we set therefore

$$\delta S \equiv S^e_{A,\text{imp+bdr}} - S^o_{A,\text{imp+bdr}}. \tag{7}$$

We note that, in this context, $\lambda$ in Hamiltonian (1) being relevant corresponds to a situation where $\delta S$ will evolve from something close to (and slightly larger than) $-\ln 2$ to $0$ as $L$ increases for a fixed $\lambda$ (that is, healing occurs). In contrast, $\lambda$ being irrelevant means $\delta S$ evolves from something close to (and slightly larger than) $-\ln 2$ to exactly $-\ln 2$ as $L$ increases for a fixed $\lambda$.[3]

As a final remark we stress that the coupling constant $g$ in eq. (2) should not be confused with the Affleck-Ludwig boundary entropy [55]. While our system indeed does involve a boundary, we are only interested in the difference between the cases of subsystems of even and odd length: both parities see the same (free) boundary conditions at the origin (the site with $j = 1$), so the corresponding term of $O(1)$ disappears in $\delta S$.

## 2.2 The relevant case

By general field theoretic arguments (see e.g. the discussion in [56]) we expect that in the limit of small $\lambda$ and large $L$, and *for the relevant case* ($g < 1$), results should have a universal dependency on the product $LT_B$. Note that this combination is small when $L$ or $\lambda$ is small, large when $L$ or $\lambda$ are large, and that, since results depend only on $LT_B$, increasing $L$ at fixed $\lambda$ in the scaling limit is like increasing $\lambda$ at fixed $L$: in other words, the long distance physics

---

[3]The different convention $\delta S = S^o - S^e$ would have avoided the minus signs but we did not make it, in order to remain consistent with earlier work.

can be expected to correspond with healing. Scaling per se only occurs formally in the limits $\lambda \to 0$, $L \to \infty$ with the product $L\lambda^{1/(1-g)}$ finite.

While this phenomenology is well understood in general, we focus here on aspects of entanglement in the presence of a boundary that have not been studied before except in the special free fermion case $J_z = 0$ [43]. Results confirming the qualitative RG picture are given below. Our numerical results are obtained by using the TeNPy package [57] where the density matrix renormalization group (DMRG) algorithm is used, details are in appendix A.

We consider the difference $\delta S$ of entanglements with the subsystem starting at the left boundary and ending in the middle of the modified bond (i.e. containing the spins $j = 1, \ldots, j = \ell$) for the cases $\ell$ even and $\ell$ odd, and we recall that we set $\delta S \equiv S^e - S^o$. The total system size is taken to be $L = Z\ell$, with $Z$ a factor taken to be $Z = 2$ unless otherwise specified.

We note now that there are many natural variants of the problem, since a priori nothing forces us to have the tunneling interaction be of the same nature than the bulk one. As long as the most relevant contribution to the tunneling interaction remains unchanged however, the RG flow in the continuum limit will be unchanged. In the Luttinger liquid language for instance, there may or may not be a Coulomb interaction between the two halves of the system: in the latter case the $J_z$ coupling would be absent. This is the situation we consider as an example in what follows, described by the Hamiltonian

$$
\begin{aligned}
H^B = &\sum_{j=1}^{\ell} \left( \sigma_j^x \sigma_{j+1}^x + \sigma_j^y \sigma_{j+1}^y + J_z \sigma_j^z \sigma_{j+1}^z \right) + \lambda \left( \sigma_\ell^x \sigma_{\ell+1}^x + \sigma_\ell^y \sigma_{\ell+1}^y \right) \\
&+ \sum_{j=\ell+1}^{L} \sigma_j^x \sigma_{j+1}^x + \sigma_j^y \sigma_{j+1}^y + J_z \sigma_j^z \sigma_{j+1}^z .
\end{aligned}
\tag{8}
$$

We see that in the case of (8), the small bond connecting the two chains only allows spin exchange, while in (1), an extra $zz$ interaction is also allowed. Since the $J_z$ term is irrelevant near the split fixed point, this modification should not affect the universal limit of our results, as we will see below.

We first give results for Hamiltonian (8) in Fig. 4. Totally identical results are obtained in the scaling limit for the Hamiltonian (1) as shown in Fig. 5. In particular, the value of $T_B$ is the same for the two curves. This coincidence is easily understood since the $\sigma_\ell^z \sigma_{\ell+1}^z$ term is irrelevant near the split fixed point [21]: while it affects corrections to scaling, it simply disappears in the limit $\lambda \to 0, \ell \to \infty$.

We note that in Fig. 5 the data corresponding to (8) is a little "fuzzy" while the two curves are slightly off for $LT_B \gtrsim 1$. This is due to the difficulty of reaching the scaling limit in the deep IR region, where values of $L$ unreachable by DMRG would, strictly speaking, be necessary (since the scaling limit is a double limit process $L \to \infty$, $\ell, L \to \infty$, $\ell/L$ finite or infinite). This is a familiar problem in the study of interacting systems. It practice, we take the small difference between the two curves in Fig. 5 as a measure of the uncertainty about the true location of the scaling curve.

Varying $Z$ does not change results much, even though of course the exact curve does, indeed, depend on $Z$ (see below for a detailed study near the fixed points). For the sake of brevity, we refrain from showing numerical results confirming this.

## 2.3 The irrelevant case

In this case we start from a small tunneling term but are driven at low-energy to the situation where the system is split. This can be seen in the fact that $LT_B$ increases at fixed $\lambda$ when increasing $L$ but increases at fixed $L$ when *decreasing* $\lambda$. Hence, large $L$ behaves like small $\lambda$,

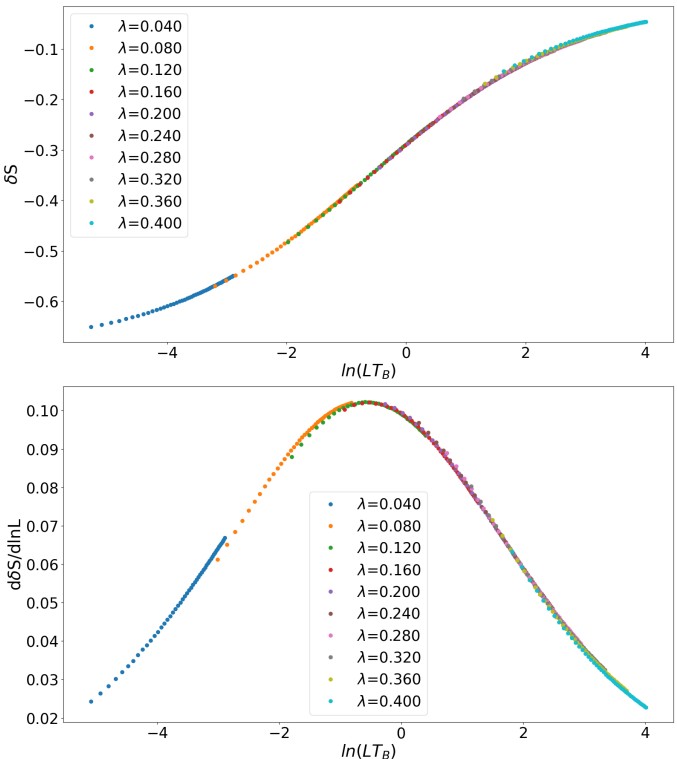

Figure 4: Healing flow for $J_z = -0.5$. Here, $Z = 2$, and the Hamiltonian is (1). The top figure shows how the difference $\delta S = S^e - S^o$, starting from (close to) $-\ln 2$ in the UV vanishes in the IR (large $LT_B$), which is expected for an homogeneous (healed) chain. The bottom figure shows the derivative of $\delta S$, which has a characteristic resonance shape.

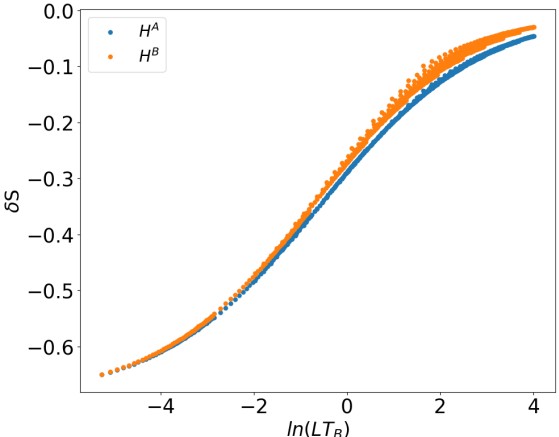

Figure 5: Healing flow for the same parameters $J_z = -0.5$ and $Z = 2$ as in figure 4. This time we compare the results obtained for Hamiltonians (1) and (8), illustrating the fact that healing occurs irrespective of the $zz$ interaction in the hopping term, and with identical universal properties. In the data shown, $\lambda$ is varied between values 0.04 and 0.4. See the comments in the text about the slight "fuzz" in the IR.

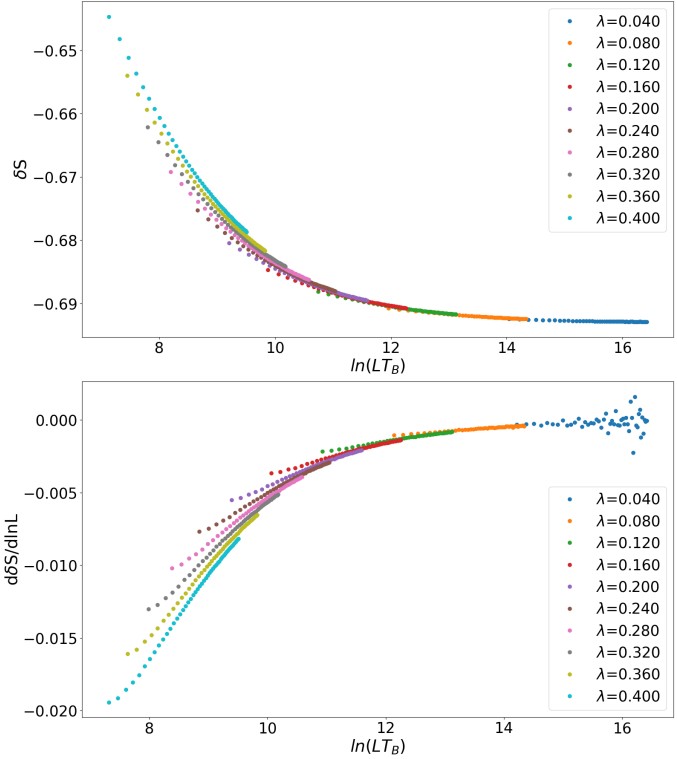

Figure 6: Flow for $J_z = 0.5$. Here, $Z = 2$, and the Hamiltonian is (8). The split fixed-point is recovered in the IR. The behavior in the UV is not universal, as expected for an irrelevant perturbation. (In particular, the healed fixed-point is not reached in the UV.)

and the split-fixed point is reached at large distances. Going to small $LT_B$ is formally equivalent to increasing $\lambda$ and thus, one would hope, to getting closer to the healed fixed-point. However, in this limit, other irrelevant operators will start playing a role, and there is no chance to reach this fixed-point without fine tuning. In practice, this simply means that the left-hand side of the curves plotting $\delta S$ as a function of $LT_B$ are *not universal* [58]. See Figs. 6 for some illustrations. In the following, we will mostly restrict to the study of relevant perturbations.

## 2.4 Some perturbative calculations

The first question to ask is how $\delta S$ varies as a function of $\lambda$ at small $\lambda$. In order to answer this question we need to think first about the situation at $\lambda = 0$, i.e. when the two systems are totally decoupled. The difference between even and odd is then spectacular. In the even case, both sides have an even number of spins and are in a (non-degenerate) ground-state of total spin $S^z = 0$ (we set $S^z = \frac{1}{2}\sum_j \sigma_j^z$). In contrast, in the odd case, both sides have a remaining spin 1/2 degree of freedom, and thus have a ground-state degenerate twice, with $S^z = \pm 1/2$. This means in particular that the shift in ground-state energy due to the presence of the $\lambda \neq 0$ term exhibits different dependencies with $\lambda$ in the even and odd cases.

### 2.4.1 The shift in energy

For the even case, this difference of ground-state energies between the $\lambda \neq 0$ and the $\lambda = 0$ cases can be obtained from non-degenerate perturbation theory and thus, by conservation of $S^z$, is quadratic in $\lambda$ at small coupling. In contrast, for the odd case, the shift comes from

degenerate perturbation theory, and since states with spin $S^z = \pm 1/2$ have the same-energy, conservation of $S^z$ does not preclude the presence of a term linear in $\lambda$. It is interesting to push these considerations a bit further by using field theoretic techniques. First, we fermionize our spin chain (we will follow standard conventions such as those in [21] whenever possible), leading to the two possible Hamiltonians

$$
\begin{aligned}
H^A = & \sum_{j=1}^{\ell} \left( c_j^\dagger c_{j+1} + \text{h.c.} + J_z (c_j^\dagger c_j - 1/2)(c_{j+1}^\dagger c_{j+1} - 1/2) \right) \\
& + \lambda \left( c_\ell^\dagger c_{\ell+1} + \text{h.c.} + J_z (c_\ell^\dagger c_\ell - 1/2)(c_{\ell+1}^\dagger c_{\ell+1} - 1/2) \right) \\
& + \sum_{j=\ell+1}^{L} \left( c_j^\dagger c_{j+1} + \text{h.c.} + J_z (c_j^\dagger c_j - 1/2)(c_{j+1}^\dagger c_{j+1} - 1/2) \right),
\end{aligned}
\tag{9}
$$

and

$$
\begin{aligned}
H^B = & \sum_{j=1}^{\ell} \left( c_j^\dagger c_{j+1} + \text{h.c.} + J_z (c_j^\dagger c_j - 1/2)(c_{j+1}^\dagger c_{j+1} - 1/2) \right) + \lambda \left( c_\ell^\dagger c_{\ell+1} + \text{h.c.} \right) \\
& + \sum_{j=\ell+1}^{L} \left( c_j^\dagger c_{j+1} + \text{h.c.} + J_z (c_j^\dagger c_j - 1/2)(c_{j+1}^\dagger c_{j+1} - 1/2) \right),
\end{aligned}
\tag{10}
$$

where $c_j^\dagger, c_j$ are the usual creation/annihilation operators with $\{c_j, c_{j'}^\dagger\} = \delta_{jj'}$. We recall the formulas for the decomposition of lattice fermions into continuous fields (see e.g. [21] for discussion of related problems)

$$
c_j \mapsto e^{iK_F j} \psi_R + e^{-iK_F j} \psi_L \,.
\tag{11}
$$

At half-filling, $K_F = \frac{\pi}{2}$ (we set the lattice spacing equal to unity). For a chain starting at $j = 1$, we have formally $c_j = 0$ for $j = 0$ to take into account the chain termination, so at that extremity the boundary conditions are $\psi_L = -\psi_R$. Meanwhile, the conditions at the other extremity depend on the parity of the length. If the last site is $l$, we set

$$
c_{l+1} \propto \psi_R + e^{-2iK_F(l+1)} \psi_L = 0 \,,
\tag{12}
$$

so at half-filling this becomes

$$
\psi_R(l+1) + (-1)^{l+1} \psi_L(l+1) = 0 \,.
\tag{13}
$$

We see that if $l$ is odd we get the same boundary conditions at $l+1$ than at 0, while if $l$ is even we get opposite boundary conditions.

Using bosonization formulas

$$
\psi_R \propto \exp(i\sqrt{4\pi}\phi_R), \ \psi_L \propto \exp(-i\sqrt{4\pi}\phi_L),
\tag{14}
$$

and handling the four-fermion term in the usual way [21] we get the continuum theory with bulk Hamiltonian

$$
H = \frac{v}{2} \int dx \left[ \Pi^2 + (\partial_x \Phi)^2 \right].
\tag{15}
$$

The compactification radius of the boson (so $\Phi$ is identified with $\Phi + 2\pi R$) is given by

$$
R \equiv \sqrt{\frac{g}{4\pi}} \,,
\tag{16}
$$

while the sound velocity is

$$v = \frac{\pi}{2} \frac{\sqrt{1-J_z^2}}{\arccos J_z}. \tag{17}$$

Note that $v = 1$ if $J_z = 0$. The boundary conditions for the bosons follow from the boundary conditions for the fermions. At the origin and in the non-interacting case, one has $\Phi = \phi_R + \phi_L = \frac{\pi}{\sqrt{4\pi}}$, which becomes $\Phi(0) = \pi R$ in general. From (13) we see that, on the right side we have $\Phi(l+1) = \pi R$ for $l$ odd and $\Phi(l+1) = 0$ for $l$ even (all these modulo $2\pi R$). So to summarize we can simply write

$$\Phi(0) = \pi R, \; \Phi(l+1) = 2\pi R S^z, \tag{18}$$

with $S^z$ integer (half an odd-integer) for $l$ even (odd).

We now consider perturbation around the almost split fixed point, with Hamiltonian $H^A$ (1) or $H^B$ (8). To all orders in perturbation theory, the correlators we need to evaluate are factorized into correlators for two decoupled sub-systems - two open chains of equal length $\ell$ in our set-up with $Z = 2$. Let us now consider the shift in ground-state energy due to the tunneling. We start with the case $\ell$ **even** where the ground-state of either half is non degenerate. The Hamiltonian we must consider is

$$\begin{aligned}
H = &\frac{v}{2} \int_0^\ell dx \left[ (\Pi^{(1)})^2 + (\partial_x \Phi^{(1)})^2 \right] + \frac{v}{2} \int_\ell^{2\ell} dx \left[ (\Pi^{(2)})^2 + (\partial_x \Phi^{(2)})^2 \right] \\
&+ \lambda Z_\lambda \cos \frac{\beta}{\sqrt{2}} (\tilde{\Phi}^{(1)}(\ell) - \tilde{\Phi}^{(2)}(\ell)),
\end{aligned} \tag{19}$$

where $Z_\lambda$ is the renormalization factor between the renormalized and the bare couplings (a thorough discussion of such factors is provided in the next section), where we have used $Z = 2$ so $L = 2\ell$, and we have set

$$\beta = \sqrt{2\pi g}. \tag{20}$$

Note that the tunneling term is expressed in terms of the *dual* field $\tilde{\Phi} = \phi_R - \phi_L$ since the field $\Phi$ takes a fixed value on either side of the tunneling bond.

The general strategy in this kind of (conformal perturbation theory) calculation is to compute partition functions and extract quantities such as the energy by looking at the leading behavior of large systems [58]. To do this, we need to introduce (imaginary) time, and thus discuss propagators in a geometry corresponding to a strip [59] (since we deal with finite systems in the space direction). Specifically, we need correlators of (exponentials of) the dual field $\tilde{\Phi}$ with Dirichlet boundary conditions, which are the same as correlators of (exponentials of) the field $\Phi$ itself with Neumann boundary conditions. Denoting by $y$ the imaginary time coordinate on a strip of width $\ell$, we find the propagator on the edge (i.e. for points on the right (or left) boundary) to be

$$\langle \tilde{\Phi}^{(i)}(y) \tilde{\Phi}^{(i)}(y') \rangle = -\frac{1}{2\pi} \ln \left| \frac{\ell}{\pi} \sinh \frac{\pi v}{\ell} (y - y') \right|^2 \tag{21}$$

(with $i = 1, 2$), so we have

$$\langle e^{i\frac{\beta}{\sqrt{2}}(\tilde{\Phi}^{(1)}(y) - \tilde{\Phi}^{(2)}(y))} e^{-i\frac{\beta}{\sqrt{2}}(\tilde{\Phi}^{(1)}(y') - \tilde{\Phi}^{(2)}(y'))} \rangle = \frac{1}{\left| \frac{\ell}{\pi} \sinh \frac{\pi v}{\ell}(y - y') \right|^{\frac{\beta^2}{\pi}}}, \tag{22}$$

where one should note the apparition of the sound velocity $v$ on the right-hand side - due to the fact that the continuum limit of the lattice Hamiltonian is not isotropic in space/(imaginary)time.

Going back to the shift of the ground state-energy, the first order correction vanishes because, in the ground-state with Dirichlet boundary conditions, $\langle e^{i\beta\tilde{\Phi}} \rangle = 0$. To get the second order, we determine the partition function in an annulus geometry with the imaginary time length $\Lambda \gg \ell$. It follows that the shift in energy is proportional, for $g < \frac{1}{2}$, to the finite integral

$$\lambda^2 \frac{Z_\lambda^2}{2} \left(\frac{\ell}{\pi}\right)^{-2g} \int_{-\infty}^{\infty} dy \frac{1}{|\sinh \frac{\pi v}{\ell} y|^{2g}} = \frac{\lambda^2 Z_\lambda^2}{v} \left(\frac{\ell}{2\pi}\right)^{1-2g} \frac{\Gamma(g)\Gamma(1-2g)}{\Gamma(1-g)} . \tag{23}$$

Since $\lambda \propto T_B^{1-g}$, this goes as $\ell^{-1}(\ell T_B)^{2-2g}$. Note that in (23) we did not worry about the UV cutoff (the fact that our system is defined on a lattice), which adds a non-universal term. This term is in fact crucial to render the integrals finite when $\frac{1}{2} < g < 1$, and the integral is UV divergent. It contributes in general

$$\frac{\lambda^2}{v} \frac{Z_\lambda^2}{2} \left(\frac{\ell}{\pi}\right)^{-2g} \left(\frac{\ell}{\pi}\right)^{2g} \int_a \frac{dy}{y^{2g}} = \frac{\lambda^2}{v} \frac{Z_\lambda^2}{2} a^{1-2g} ,$$

and thus (choosing origins of energies so that the decoupled system has vanishing energy) we obtain the change in energy due to the modified bond

$$E^e \approx \lambda^2 \left(c_1 \ell^{1-2g} + c_2\right) , \tag{24}$$

where $c_1$ is a universal constant, while $c_2$ is not.

In the **odd** case the ground state is degenerate four times, since each half has a leftover spin 1/2. Ground states can be written symbolically (see below eq. (38) $|\Omega\rangle_{\alpha\beta} = |\alpha\rangle \otimes |\beta\rangle$, with $\alpha, \beta = \pm 1$. In each of the subsystems, the raising/lowering spin at the extremity has non-zero matrix elements between $|\pm\alpha\rangle$. Carrying out degenerate perturbation theory we expect a shift in energy proportional to $\lambda$, whereas the non-zero matrix elements should scale as $L^{-g}$ by dimensional analysis. Hence in this case

$$E^o \approx \lambda \left(c_3 \ell^{-g} + c_4 \lambda\right) , \tag{25}$$

where we added a pure $\lambda^2$ correction just like in the even case.

In conclusion, we see that the behaviors of the energy for even and odd cases are quite different, with, in the scaling limit,

$$\begin{aligned} E^e &\propto \ell^{-1}(\ell T_B)^{2-2g} , \\ E^o &\propto \ell^{-1}(\ell T_B)^{1-g} . \end{aligned} \tag{26}$$

These results have been checked numerically - an example is provided on figure 7.

### 2.4.2 The entanglement

The full perturbative computation of the entanglement near the split fixed point would involve some technical aspects that are best discussed elsewhere. We shall content ourselves with carrying out a schematic analysis of the problem, which will reveal nevertheless the most important facts.

We start by considering the simplest case $Z = 2$ and $\ell$ even (recall that $\ell$ is the length of the subsystem and $L = Z\ell$). When $\lambda = 0$, the subsystem and its complement are both in a ground-state which mimics the case $\ell = 2$: the spin on either side of the cut is up or down, and since the total $S^z$ for each subsystem vanishes, the remaining $\ell - 1$ spins have a total $S^z$ that is down or up. In other words, the ground state of each subsystem can be written

$$|0\rangle = \frac{|(+)-\rangle - |(-)+\rangle}{\sqrt{2}} , \tag{27}$$

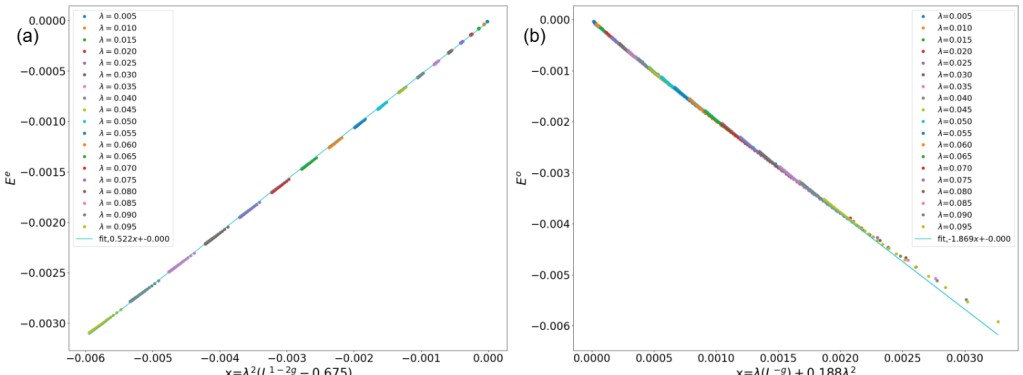

Figure 7: Study of the change in energy $E^{e/o}=E^{e/o}(\lambda)$-$E^{e/o}(\lambda = 0)$ at small $\lambda$, $J_z = -0.3$, for Hamiltonian (1). (a) Even case (b) Odd case. Observe how the two parities give rise to different dependencies.

where $(+)$ and $(-)$ stand for the state of the remaining $\ell-1$ spins with this total magnetization (in this section in general, the terms within parenthesis in bras or kets will be short-hand descriptions of the remaining $\ell-1$ spins on both sides of the modified bond). By $Z_2$ symmetry, $(-)$ is obtained from $(+)$ by flipping all spins. The ground state for the whole system is then $|\Omega\rangle = |0\rangle \otimes |0\rangle$.

Imagine now calculating the ground state when $\lambda \neq 0$ is small by using perturbation theory. Restrict for simplicity to the case of the Hamiltonian $H^B$. The perturbation $\lambda V \equiv \frac{\lambda}{2}\left(\sigma_\ell^+ \sigma_{\ell+1}^- + h.c.\right)$ acting only on the extremity spins of the two subsystems couples to eigenstates of the decoupled system where one side has spin one and the other spin minus one. Call the corresponding eigenstates $|1_n\rangle$ and $|-1_n\rangle$, with energy $E_n$. Similarly, two insertions of $V$ couple to eigenstates where both sides have vanishing spin. Call the corresponding eigenstates $|0_n\rangle$ (with $|0_0\rangle = |0\rangle$). To second order we have then

$$|\Omega\rangle = |0\rangle \otimes |0\rangle + \lambda \sum_{n,m} a_{nm}\left(|1_n\rangle \otimes |-1_m\rangle + |-1_n\rangle \otimes |1_m\rangle\right) + \lambda^2 \sum_n b_n\left(|0_n\rangle \otimes |0\rangle + |0\rangle \otimes |0_n\rangle\right), \quad (28)$$

where from first order perturbation theory,[4]

$$a_{nm} = \frac{1}{2} \frac{\langle 1_n|(+)+\rangle\langle -1_m|(-)-\rangle}{(E_n - E_0)(E_m - E_0)}. \quad (29)$$

Taking the trace on the second subspace of the full density operator $\rho = |\Omega\rangle\langle\Omega|$ gives the contribution to the reduced density operator for one half of the system

$$\rho_A \equiv \text{Tr}_B\rho = (1 + \lambda^2(b_0 + b_0^*))|0\rangle\langle 0| + \lambda^2 \sum_{nm} A_{nm}\left(|1_n\rangle\langle 1_m| + |-1_n\rangle\langle -1_m|\right)$$
$$+ \lambda^2 \sum_{n>0} b_n|0_n\rangle\langle 0| + b_n^*|0\rangle\langle 0_n|, \quad (30)$$

where $A_{nm} = \sum_p a_{np}a_{pm}^*$.

The reduced density matrix is thus the sum of three operators acting on different subspaces, and whose products are all vanishing. Symbolically we write

$$\rho_A = \text{Tr}_B\rho = (1 + \lambda^2(b_0 + b_0^*))|0\rangle\langle 0| + \lambda^2 A + \lambda^2 B, \quad (31)$$

---

[4]We do not use this exact form below, and only give it to illustrate the point.

so that the ratio [60]

$$R_p \equiv \frac{\text{Tr } \rho_A^P}{(\text{Tr } \rho_A)^p} \,, \tag{32}$$

has the structure

$$R_p = \frac{(1 + \lambda^2 (b_0 + b_0^*))^p + \lambda^{2p} (\text{Tr} A^p + \text{Tr} B^p)}{(1 + \lambda^2 (b_0 + b_0)^* + \lambda^2 (\text{Tr} A + \text{Tr} B))^p} \,. \tag{33}$$

By standard arguments $R_p$ is the Renyi entropy of order $p$ and we can obtain from it the (Von Neumann) entanglement entropy[5] as $S = -\frac{d}{dp} R_P \big|_{p=1}$. Based on the foregoing schematic calculation we get

$$S = -\frac{d}{dp} R_P \bigg|_{p=1} = -2|X|^2 \lambda^2 \left( -\frac{1}{2} + \ln |X|^2 \lambda^2 \right) , \tag{34}$$

where $X$ is a coefficient that would follow from an explicit calculation of the coefficients $b_n, A_{nm}$ above. Our purpose here is not to obtain quantitative results but only the general perturbative structure of $S$. We thus contend ourselves by concluding from this series of arguments that the entanglement entropy in the even case has a leading term going as $\lambda^2 \ln \lambda$.

It is now useful to summarize the foregoing discussion in more general terms. In the even case,the ground state of each of the two decoupled systems is non-degenerate and has spin $S^z = 0$. Since the full Hamiltonian commutes with the total spin

$$[H, S_A^z + S_B^z] = 0 \,, \tag{35}$$

the reduced density matrix $\rho_A$ commutes with the spin $S_A^z$ [61]:

$$[\rho_A, S_A^z] = 0 \,. \tag{36}$$

Right at the decoupled point, $\rho_A$ only has matrix elements between the factorized ground-state and itself, both at $S_A^z = 0$. However, under the tunneling perturbation, the new ground-state acquires components onto states which, while having total spin $S_A^z + S_B^z = 0$, have $S_A^z = \pm 1$. After tracing over the $B$ degrees of freedom, and writing $\rho_A$ in block diagonal form with blocks labelled by $S_A^z$, this means that, two second order in perturbation theory, we have the structure

$$\rho_A = \text{Tr}_B \rho = \begin{pmatrix} \boxed{\rho^{(0)}} & 0 & 0 & \cdots \\ 0 & \boxed{\rho^{(1)}} & 0 & \cdots \\ 0 & 0 & \boxed{\rho^{(-1)}} & \cdots \\ \cdots & \cdots & \cdots & \cdots \end{pmatrix} , \tag{37}$$

where the superscript refer to values of $S_A^z$ and the dots on the diagonal contain blocks of higher charge. The crucial point now is that $\rho^{(\pm n)}$ for $n \neq 0$ is a contribution of order $\lambda^{2n}$ since it takes $n$ actions of the perturbation to produce a state with spin $S^z = \pm n$ starting from a state of vanishing spin. We thus see immediately that we can expect a structure as in (37) and consequently, after calculating the Reny entropy and taking the limit $p \to 1$, generate the leading term $\lambda^2 \ln \lambda$. Note meanwhile that, were we to carry out the perturbative expansion to higher orders, only terms *even* in $\lambda$ would be encountered.

We now consider the case $\ell$ odd. Things are then a bit different. Exactly at $\lambda = 0$ there is a potential ambiguity since the left and right hand sides are now both degenerate twice.

---

[5]We use here the formulation of the Von Neumann entropy as the derivative of the Renyi entropies as a function of $n$ as $n \to 1$. This is of course identical with the maybe more familiar definition $S = -\ln \frac{1}{p-1} \ln R_p \big|_{p=1}$.

The entanglement is not even defined at this point without specification of the state of the full system. However, as soon as $\lambda \neq 0$, this degeneracy is broken, and the ground state becomes unique and has $S^z = 0$. It is easy to identify this state in the case $Z = 2$, where the system with or without perturbation is symmetric under exchanges of the two sides and conserves the total spin: the ground state at finite $\lambda$ remains in the sector antisymmetric under the exchange, and with $S^z = 0$.

When $\lambda = 0$ we can write the (normalized) ground states of each side as combinations

$$
\begin{aligned}
|+\rangle &= \lambda_+^+ |(0)+\rangle + \lambda_-^+ |(++)-\rangle , \\
|-\rangle &= \lambda_-^- |(0)-\rangle + \lambda_+^- |(--)+\rangle ,
\end{aligned}
\tag{38}
$$

where once again $(0), (++), (--)$ stand for states of the remaining $\ell-1$ (now, an even number) spins. We then choose the ground state of the whole system to be

$$
|\Omega\rangle = \frac{|+-\rangle - |-+\rangle}{\sqrt{2}} .
\tag{39}
$$

The density matrix of the lhs then reads schematically, in the basis (38)

$$
\mathrm{Tr}_B \rho = \frac{1}{2}
\begin{pmatrix}
(\lambda_+^+)^2 & \lambda_+^+ \lambda_-^+ & 0 & 0 \\
\lambda_+^+ \lambda_-^+ & (\lambda_-^+)^2 & 0 & 0 \\
0 & 0 & (\lambda_+^-)^2 & \lambda_+^- \lambda_-^- \\
0 & 0 & \lambda_+^- \lambda_-^- & (\lambda_-^-)^2
\end{pmatrix} .
\tag{40}
$$

Using that the ground states (38) are normalized we can easily calculate from this the Reni entropy and show that it gives, as expected, rise to $S = \ln 2$.

Now going through the same charge conservation arguments as before, we see that, when perturbing the ground state we will have to carry out a calculation similar to the one of the even case, resulting in a charge resolved structure for the density matrix now of the type

$$
\mathrm{Tr}_B \rho =
\begin{pmatrix}
\boxed{\rho^{(1/2)}} & 0 & 0 & 0 & \cdots \\
0 & \boxed{\rho^{(-1/2)}} & 0 & 0 & \cdots \\
0 & 0 & \boxed{\rho^{(3/2))}} & 0 & \cdots \\
0 & 0 & 0 & \boxed{\rho^{(-3/2))}} & \cdots \\
\cdots & \cdots & \cdots & \cdots & \cdots
\end{pmatrix} ,
\tag{41}
$$

where the terms $\rho^{(\frac{1}{2}+n)}$ will come with factors $\lambda^{2n}$. Once again we will get in the end terms that are even in $\lambda$, with a leading correction going as $\lambda^2 \ln \lambda^2$.

We thus conclude from this discussion that both in the even and odd cases the entanglement entropy varies at small coupling as

$$
S^{e,o} \approx -k^{e,o} \lambda^2 \ln(\mathrm{const}\, |\lambda|) .
\tag{42}
$$

Note how different this is from the behavior of the energy, where the leading behaviors at small $\lambda$ were given respectively by (24) and (25). Comparing energy shifts on the even and odd cases would not have made much sense since they have a different functional form, in contrast with the case of the entanglements.

Numerical results fully confirm the results in (42) (see figure 8). We also give below a determination of the slopes of the leading terms, although we do not have a full analytical derivation at this stage. To conclude, we see that, in contrast with the energy, the entanglement at small $\lambda$ behaves similarly (but not identically) in the even and odd cases.

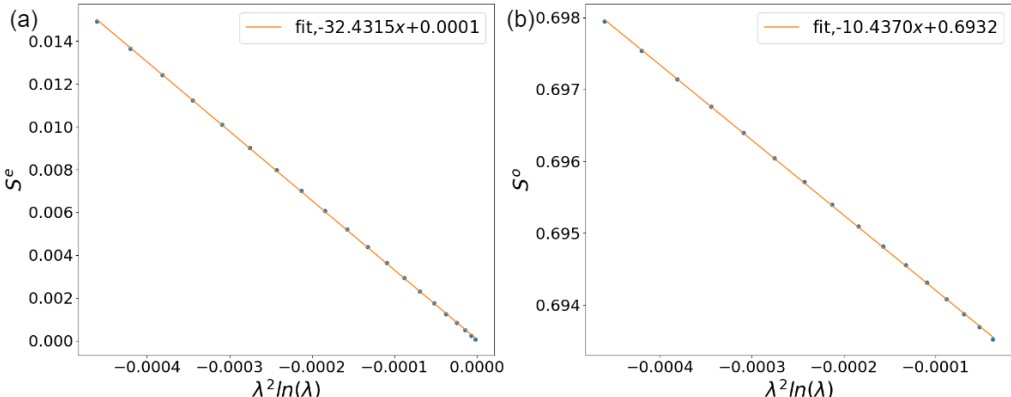

Figure 8: Study of the entanglement at small $\lambda$, $J_z$=-0.5, L=400 for Hamiltonian (1), confirming the *same* dependency $S \propto -\lambda^2 \ln|\lambda|$. (a) Even case. (b) Odd case.

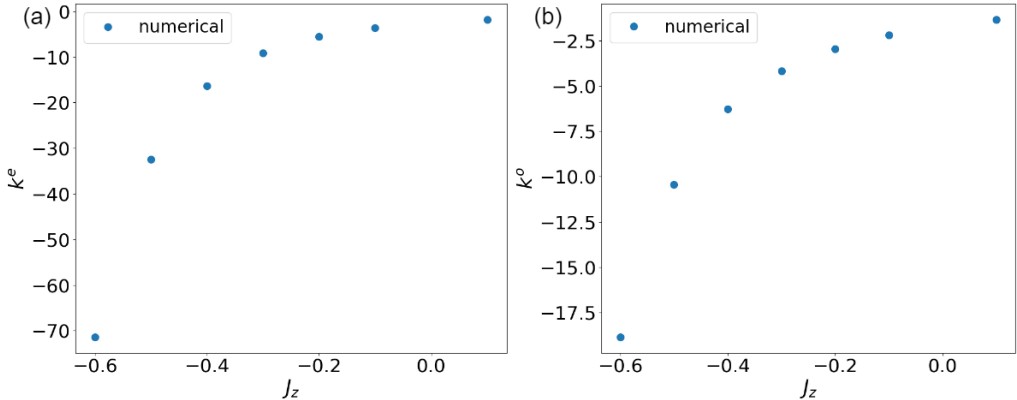

Figure 9: The slopes of the $\lambda^2 \ln|\lambda|$ leading term as a function of $J_z$ for Hamiltonian (1). (a) Even case. (b) Odd case.

# 3 Physics around the homogeneous fixed-point

Like in the previous section, we discuss some generalities and numerical results. We then launch into a full perturbative calculation[6] of $\delta S$, and a detailed comparison with our numerics. This involves non-trivial renormalization of the coupling constant.

## 3.1 Generalities

We can also consider the vicinity of the homogeneous (uniform) fixed point. Here again, several natural choices of Hamiltonians are possible. We can in particular imagine a weakened interaction between sites $\ell, \ell + 1$, or, in the Luttinger liquid version, a situation where tunneling is weakened while the Coulomb interaction between the two halves remains unaffected.

---

[6]This case is simpler than the vicinity of the split fixed point, as it does not involve degeneracies in the unperturbed ground-state.

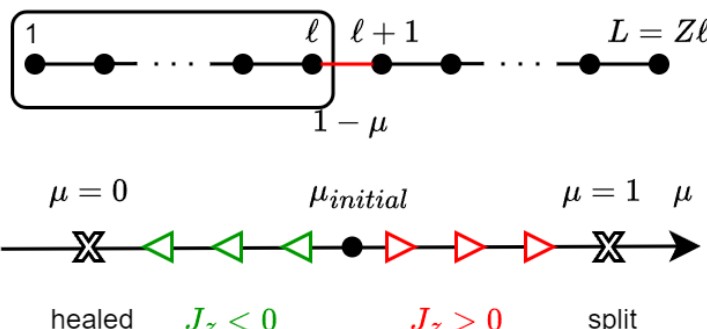

Figure 10: The different types of flows near the homogeneous fixed-point ($\mu$ small). Compare with figure 2.

These are the two situations we shall study in more detail, with corresponding Hamiltonians

$$H^A = \sum_{j=0}^{\ell} \left( \sigma_j^x \sigma_{j+1}^x + \sigma_j^y \sigma_{j+1}^y + J_z \sigma_j^z \sigma_{j+1}^z \right) + (1-\mu)\left( \sigma_\ell^x \sigma_{\ell+1}^x + \sigma_\ell^y \sigma_{\ell+1}^y + J_z \sigma_\ell^z \sigma_{\ell+1}^z \right)$$

$$+ \sum_{j=\ell+1}^{L} \sigma_j^x \sigma_{j+1}^x + \sigma_j^y \sigma_{j+1}^y + J_z \sigma_j^z \sigma_{j+1}^z, \tag{43}$$

and

$$H^B = \sum_{j=0}^{\ell} \left( \sigma_j^x \sigma_{j+1}^x + \sigma_j^y \sigma_{j+1}^y + J_z \sigma_j^z \sigma_{j+1}^z \right) + (1-\mu)\left( \sigma_\ell^x \sigma_{\ell+1}^x + \sigma_\ell^y \sigma_{\ell+1}^y \right) + J_z \sigma_\ell^z \sigma_{\ell+1}^z$$

$$+ \sum_{j=\ell+1}^{L} \sigma_j^x \sigma_{j+1}^x + \sigma_j^y \sigma_{j+1}^y + J_z \sigma_j^z \sigma_{j+1}^z. \tag{44}$$

Perturbing the coupling near the uniform fixed point corresponds in the continuum limit to coupling an operator of dimension $(\text{length})^{-g^{-1}}$ (together with an operator of dimension $(\text{length})^{-2}$, see below). We see that the regions of relevance and irrelevance are switched with respect to the previous section, and that

$$\text{near the homogeneous fixed point } \mu \text{ is irrelevant if } J_z < 0,$$

$$\text{near the homogeneous fixed point } \mu \text{ is relevant if } J_z > 0. \tag{45}$$

The corresponding flows are sketched in Fig. 10. We see now that

$$\dim \mu = (\text{length})^{g^{-1}-1}, \tag{46}$$

so, using the same kind of scaling argument as for the case of an almost split chain, we now expect the properties to have a universal dependency on $L\Theta_B$ with

$$\Theta_B \equiv \mu^{1/(1-g^{-1})}. \tag{47}$$

## 3.2 Numerics for the entanglement

In the relevant case, we now flow from the homogeneous to the split fixed point - this is the standard situation in the Kane-Fisher problem, where a slightly diminished bond in the chain gives rise at large distance (small energies) to the same behavior as a chain split in half. As a result, $\delta S$, which vanishes in the homogeneous case flows to $-\ln 2$, as is illustrated in Fig. 11 for two different values of the coupling $J_z$.

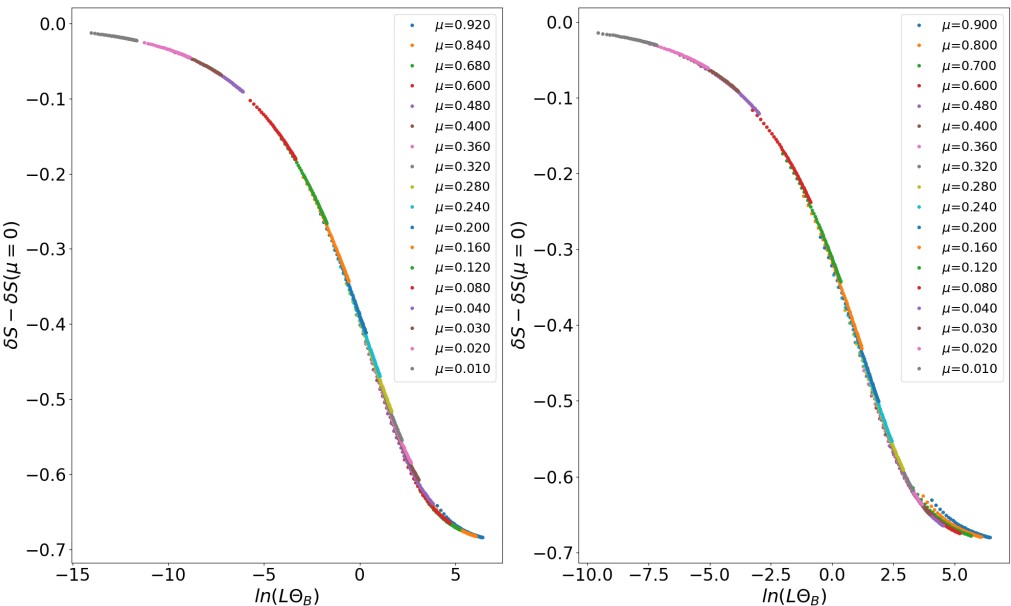

Figure 11: Flow for $J_z = 0.5, 0.7$. Here, $Z = 2$, and the Hamiltonian is (43). The weakly perturbed chain flows to the split fixed-point in the IR.

## 3.3 Some perturbative calculations

We now study some feature of the entanglement in perturbation theory. To carry out the calculation, we once again turn to bosonization. This time, since we start with a homogeneous chain, we have to consider a single bosonic theory on the half-line (or a segment of length $Z\ell$ if the system is finite), instead of two separate systems. We start with (44) and need the crucial bosonization formula [50]

$$\sigma_j^x \sigma_{j+1}^x + \sigma_j^y \sigma_{j+1}^y = 2(-1)^j c_1^\pm \cos\frac{\Phi}{R}(x=j), \qquad (48)$$

where $c_1^\pm$ is a constant equal to $\frac{1}{\pi}$ in the non-interacting case, but otherwise not known exactly (see below). Note we only represented the leading term, which is of dimension $\frac{1}{4\pi R^2} = g^{-1}$. The next term would be proportional to $\left(\frac{\partial \Phi}{\partial x}\right)^2$, and thus is of dimension 2: it becomes in fact the most relevant one when $g^{-1} = 2$, that is $J_z = -\frac{\sqrt{2}}{2}$ [62]. We will not study this region, and restrict therefore to $g > \frac{1}{2}$ or $J_z > -\frac{\sqrt{2}}{2}$. In the non-interacting case $g = 1$, $R = \sqrt{4\pi}$ and we shall recover the results in [43].

The Hamiltonian corresponding to (44) is then

$$H = \frac{v}{2} \int_0^\infty dx \left[\Pi^2 + (\partial_x \Phi)^2\right] - \frac{\mu_R}{\pi}(-1)^\ell \cos\frac{\Phi}{R}(x=\ell). \qquad (49)$$

Note that in (49) we have introduced a *renormalized coupling constant* $\mu_R$. Indeed, while in the non-interacting case $\mu_R = \mu$ as defined with the lattice Hamiltonian (44) (which is the same as (43) in this case), in the presence of interactions, renormalization effects lead to $\mu_R = Z_\mu \mu$. The constant $Z_\mu$ - essentially the proportionality constant $c_1^\pm$ in (48) - is not universal. Its value for the XXZ spin chain is not known exactly, but has been determined numerically to a great accuracy in [63] (for earlier work see [64–66]). We will use the values deduced from Table I in [63], after the correspondence $Z_\mu^B = \sqrt{8\pi^2 B_1^\pm} = \sqrt{4\pi^2} c_1^\pm$, while $J_z = \Delta$ (so that $Z_\mu^B = 1$

for $J_z = 0$). In fact, it will turn out that the relevant quantity is the ratio $\frac{Z_\mu^B}{v}$, where $v$ is given in (17)

For the Hamiltonian (43), we need one more bosonization formula to complement (48)

$$\sigma_j^z \sigma_{j+1}^z = c_1^z (-1)^j \cos \frac{\Phi(x=j)}{R} \tag{50}$$

(where again this holds to leading order and for for $J_z > -\frac{\sqrt{2}}{2}$). In this region, the Hamiltonian in the continuum limit if still (49) but now with a different value of the renormalization constant $Z_\mu^A$, since

$$\sigma_j^x \sigma_{j+1}^x + \sigma_j^y \sigma_{j+1}^y + J_z \sigma_j^z \sigma_{j+1}^z = (-1)^j (2c_1^\pm + c_1^z) \cos \frac{\Phi}{R}(x=j). \tag{51}$$

It follows that the new renormalization constant is $Z_\mu^A = \sqrt{8\pi^2} \left( \sqrt{B_1^\pm} + \frac{J_z}{2} \sqrt{B_1^z} \right)$. After dividing by $v$, this gives rise to the values listed in the tables below.

We now use these results to study the entanglement. The strategy is the same as the one used in [43],[7] but for the convenience of the reader we repeat the important steps.

Since the boson sees Dirichlet boundary conditions, we have the non-trivial one-point function in the half-plane,

$$\langle e^{i\frac{\Phi}{R}(x)} \rangle_{\text{HP}} = \frac{1}{(2x)^{2h}}. \tag{52}$$

We now consider the Rényi-entropy of the interval $(0, \ell)$ using the replica approach of [60]. We start by taking $L = \infty$: the case of finite $L$ can then be handled using conformal transformations. The calculation involves a $p$-sheeted Riemann surface $R_p$ (with $p$ an integer) with a cut extending from the origin to the point of coordinates $(x = \ell, \tau = 0)$, where we use $\tau$ to denote (imaginary) time coordinates. We expand the partition function, and thus need for this the one point function of $e^{i\frac{\Phi}{R}}$ on the corresponding surface. We obtain it by uniformizing via two mappings: If $w$ is the coordinate on $R_p$, the map

$$u = -\left( \frac{w - i\ell}{w + i\ell} \right)^{1/p}, \tag{53}$$

maps onto the disk $|u| \le 1$, while the map

$$z = -i \frac{u-1}{u+1}, \tag{54}$$

maps the disk onto the ordinary half-plane. Using Eq. (52), we obtain

$$\langle e^{i\frac{\Phi}{R}(w,\bar{w})} \rangle_{R_p} = \left( \frac{2\ell}{p} \right)^{2h} \frac{[(w-i\ell)(w+i\ell)(\bar{w}-i\ell)(\bar{w}+i\ell)]^{h(\frac{1}{p}-1)}}{\left[ (w+i\ell)^{1/p}(\bar{w}-i\ell)^{1/p} - (w-i\ell)^{1/p}(\bar{w}+i\ell)^{1/p} \right]^{2h}}. \tag{55}$$

Here, we have used the fact that vertex operators are primary fields, and thus transform without any anomalous terms. Since two transformations are in fact required to map $R_p$ onto the half-plane $w \mapsto u \mapsto z$, we have refrained from writing the intermediate steps, and only given the final result.

Specializing to the perturbation with coordinates $(x = \ell, \tau)$ we get

$$\langle \cos \frac{\Phi}{R}(w, \bar{w}) \rangle_{R_p} = \left( \frac{2\ell}{p} \right)^{2h} \frac{[v^2 \tau^2 [v^2 \tau^2 + 4\ell^2]^{h(\frac{1}{p}-1)}}{[(v^2 \tau^2 + 4\ell^2)^{1/p} - (v\tau)^{2/p}]^{2h}}. \tag{56}$$

---

[7]What is called $(\lambda - 1)$ in eq. 43 in this reference is called $\mu$ here, while in eq. 44 of this reference, we have $\beta = \frac{1}{R}$, so $\frac{\beta^2}{4\pi} = g^{-1}$ and $h = \frac{1}{2} g^{-1}$.

The asymptotic behaviors at large distance are now $\langle\cos\frac{\Phi}{R}(w,\bar{w})\rangle_{R_p} \approx (2\ell)^{-2h}$, $\tau \gg \ell$ and $\langle\cos\frac{\Phi}{R}(w,\bar{w})\rangle_{R_p} \approx \left[\frac{1}{p}(2\ell)^{-1/p}\tau^{\frac{1}{p}-1}\right]^{2h}$, $\tau \ll \ell$. Like in [43] we calculate perturbatively the partition function of the system on $R_p$ (for $\ell$ even):

$$R_p(\mu_R) \equiv \frac{Z_p}{(Z_1)^p} = \frac{\int_{\text{twist}}[\mathcal{D}\phi_1]\dots[\mathcal{D}\phi_p]\exp\left\{-\sum_{i=1}^p\left(A[\phi]_i - \frac{\mu}{\pi}\int d\tau_i\cos\beta\phi_i(\ell,\tau_i)\right)\right\}}{\left(\int[\mathcal{D}\phi]\exp\{-\left(A[\phi] - \frac{1}{\pi}\mu\int d\tau\cos\beta\phi(\ell,\tau)\right)\}\right)^p}, \quad (57)$$

and the integral in the numerator is taken with the sewing conditions

$$\phi_i(0 \le x \le \ell, \tau = 0^+) = \phi_{i+1}(0 \le x \le \ell, \tau = 0^-). \quad (58)$$

As usual, we trade this problem of $p$ copies of the field for a single field on $R_p$ [60]. With no modified bond, we get the known result,

$$R_p(\mu_R = 0) \propto \left(\frac{1}{\ell}\right)^{2h_p}, \qquad h_p = \frac{c}{24}\left(p - \frac{1}{p}\right), \quad (59)$$

so

$$S = -\left.\frac{d}{d_p}R_p\right|_{p=1} = \frac{1}{6}\ln\frac{\ell}{p} \quad (60)$$

(recall we set the lattice spacing (the UV cutoff) equal to one). This is the usual entanglement entropy near a boundary - and *we have discarded terms of order 1 which are independent of $\mu$.*

We now proceed to calculate the ratios $R_p(\mu_R)/R_p(\mu_R = 0)$. To leading order in $|\mu_R| \ll 1$, one finds

$$\frac{R_p(\mu_R)}{R_p(\mu_R = 0)} = 1 + \frac{\mu_R}{\pi}\left(\sum_{i=1}^p\int d\tau_i\langle\cos\beta\phi(\ell,\tau_i)\rangle_{R_p} - p\int d\tau\langle\cos\frac{\Phi}{R}(\ell,\tau)\rangle_{\text{HP}}\right). \quad (61)$$

Here, $\tau_i$ parametrizes the $p$ copies (on the $p$ sheets of $R_p$) of the line along which the perturbation is applied (in the Euclidian formulation).

Like we observed in [43] in the case $h = \frac{1}{2}$, the resulting integral is convergent - in fact, thanks to the subtraction coming from the denominator, it turns out to be *always convergent*, even when the perturbation is irrelevant. Setting $v\tau = 2\ell\tan\theta$ we have[8]

$$\frac{R_p(\mu_R)}{R_p(\mu_R = 0)} = 1 + \frac{2\mu_R}{\pi v}(2\ell)^{1-2h}\int_0^{\pi/2}\frac{d\theta}{\cos^2\theta}\left[p^{1-2h}\frac{(\sin\theta)^{2h(\frac{1}{p}-1)}\cos^{4h}\theta}{[1-(\sin\theta)^{2/p}]^{2h}} - p\right]$$
$$\equiv 1 + \frac{2\mu_R}{v\pi}(2\ell)^{1-2h}I_p, \quad (62)$$

where the second equation defines the integral $I_p$. Like in [43] we get the correction to the entanglement by

$$S = \frac{1}{6}\ln(\ell) + \frac{2\mu_R}{\pi v}(2\ell)^{1-2h}\left.\frac{d}{dp}I_p\right|_{p=1}. \quad (63)$$

Remarkably, the resulting integral differs from the one when $h = \frac{1}{2}$ by a simple factor: $\left.\frac{d}{dp}I_p\right|_{p=1}(h) = 2h\left.\frac{d}{dp}I_p\right|_{p=1}(h = \frac{1}{2})$,[9] and we find in the end the simple result

$$S = \frac{1}{6}\ln\left(\frac{\ell}{a}\right) + \frac{1}{3}g^{-1}\frac{\mu_R}{v}(2\ell)^{1-g^{-1}} + O(\mu_R^2), \quad (64)$$

---

[8]We note there is an unfortunate typo in eq. 54 of [43]: two of the cosines in the bracket, should be sinuses, as can be seen by setting $h = \frac{1}{2}$ in (62). Also, notice that $\mu$ in [43] is equal to $\frac{\mu_R}{\pi}$ in the present paper.

[9]It converges at $\theta = \frac{\pi}{2}$ in all cases.

where we used that $2h = g^{-1}$ and $\frac{d}{dp}I_p\big|_{p=1}(h = \frac{1}{2}) = \frac{\pi}{6}$. As in [43] the result only holds for $L = \infty$. When the ratio $\frac{\ell}{L}$ is finite, finite size effects have to be taken into account - see below and [43] for the case $Z = 2$.

Comparing odd and even cases amounts to changing the sign of $\mu$ as discussed in [43]. This leads finally to our main result

$$\delta S = \frac{2}{3}g^{-1}\frac{\mu_R}{\nu}(2\ell)^{1-g^{-1}} + O(\mu_R^2). \tag{65}$$

In the non-interacting case $J_z = 0$ we have $g = 1$ and we find $\delta S = \frac{2}{3}\mu$ like in eq. 58 of [43]. When the system has finite size $L = 2\ell$ (so $Z = 2$), we find, generalizing eq. 60 of [43] for $g = 1$ (and $\mu_R = \mu$)[10]

$$\boxed{\begin{aligned}\delta S_{Z=2} &= .636779\ g^{-1}\frac{\mu_R}{\nu}\left(\frac{4\ell}{\pi}\right)^{1-g^{-1}}\\&= .636779\ g^{-1}\frac{Z_\mu}{\nu}\mu\left(\frac{4\ell}{\pi}\right)^{1-g^{-1}}\end{aligned}}$$

(We have put a box around this equation as we consider it the most important result of this paper). As mentioned above we now observe that the integrals encountered in this calculation are *always convergent*, irrespective of the relevance of the perturbation. It follows that (66) should hold as well when the perturbation is irrelevant, i.e. when $J_z > 0$. The numerics indeed do not see anything happening when $J_z = 0$ is crossed. This is rather unexpected, since one generally expects that perturbation by an irrelevant operator leads to IR divergences. This issue may have to do with the behavior of other quantities such as the screening cloud in the Kondo problem [31, 32].

On the other hand, we emphasize that result (66) only makes sense when the hopping term is the leading (ir)relevant operator. As $J_z$ crosses the value $-\frac{\sqrt{2}}{2}$, the term of ($J_z$ independent) dimension 2 dominates, and thus (66) ceases to be valid.

## 3.4 Comparison with numerics

The numerical analysis is a little tricky because, even in the absence of a local perturbation, the entanglement is known to already exhibit an alternating dependency upon $\ell$ (that is, even odd effects that decrease with $\ell$) [67–69], leading to

$$\delta S_{Z=2}(\mu = 0) = a(g)l^{-g^{-1}}. \tag{66}$$

This correction is well identified in the literature, and the exponent usually written as $K$, the Luttinger constant, with $K = \frac{\pi}{2(\pi - \arccos J_z)} = \frac{1}{g}$. It is due to the leading irrelevant *bulk* oscillating term in the chain. We have first checked the result (66), as illustrated in Fig. 12 (a).

To leading order, we expect the correction (66) and the correction induced by the $\mu \neq 0$ perturbation to simply add up, so we can get rid of decaying oscillating terms of [67–69] by considering the difference

$$\delta S_{Z=2}(\mu) - \delta S_{Z=2}(\mu = 0) = .636779\ g^{-1}\frac{Z_\mu}{\nu}\mu\left(\frac{4\ell}{\pi}\right)^{1-g^{-1}}. \tag{67}$$

---

[10]The substitution for the $\ell$ dependent factor is $2\ell \to \frac{2L}{\pi}\sin\frac{\pi\ell}{L}$, so $2\ell \to \frac{4\ell}{\pi}$ when $L = 2\ell$. Otherwise, finite size gives rise to the same modified integral $I_p$ as in [43].

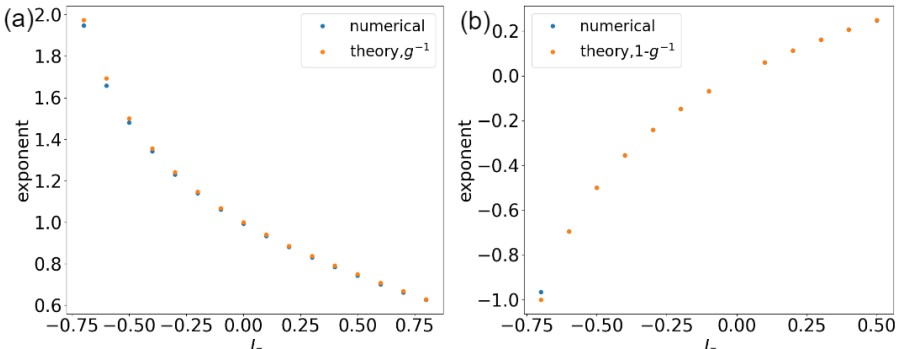

Figure 12: (a) Study of the exponent controlling the decay of $\delta S$ for a homogeneous chain as in(66). (b) Study of the exponent controlling the decay of $\delta S$ for a small perturbation of the homogeneous fixed point (67). In both cases, the numerics reproduces well the expected result.

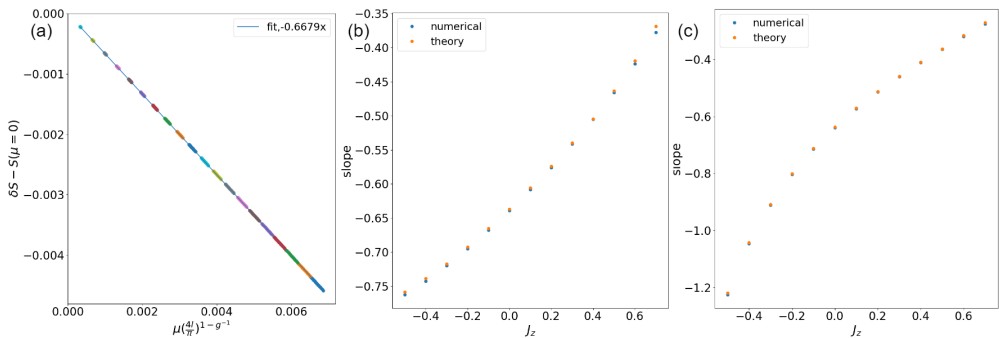

Figure 13: (a) Example of fit of $\delta S_{Z=2}(\mu) - \delta S_{Z=2}(\mu = 0)$ against $\mu \left( \frac{4\ell}{\pi} \right)^{1-g^{-1}}$ when $J_z$=-0.1 for Hamiltonian (43). The linear behavior expected from (67) is well observed, and the slope can be measured accurately. (b). Study of the slope of $\delta S$ near the homogeneous fixed point obtained by fitting (67) (see figure 13(a) for Hamiltonian (43). Blue points are numerical. Orange points are obtained from the values given in [63](c) Same as (b) but for the Hamiltonian (44).

We have studied the left hand side in what follows. Measures of the exponent obtained by plotting $\ln \left[ \delta S_{Z=2}(\mu) - \delta S_{Z=2}(\mu = 0) \right] - \ln \ell$ for small values of $\mu$ give excellent, $\mu$ independent results, as illustrated in Fig. 12(b): To obtain results for the slope itself, we fit $\delta S_{Z=2}(\mu) - \delta S_{Z=2}(\mu = 0)$ for a series of values of $J_z$ - it turns out the relevant region involves values of $\mu$ as small as $5 \, 10^{-4}$. An example of such fit is given in Fig. 13(a).

The resulting slopes are then compared with the best known numerical values in Table 1 and in Figs. 13(b) and 13(c) for the two possible Hamiltonians. Note the excellent agreement both in the relevant and irrelevant case as long as $J_z$ is not too close to $\pm 1$. As expected on general grounds due to the presence of almost marginal operators as $J_z$ approaches 1, the quality of the numerical estimates decreases in this limit.[11] As for the region of large negative $J_z$, we remind the reader that our calculation only considers the region $-\frac{\sqrt{2}}{2} < J_z$ since beyond this lower bound, the modified bond corresponds to another leading operator in the scaling limit.

---

[11]We note that the question of corrections to scaling to entanglement entropies in general is complex, especially near when an operator becomes marginal. See [69].

Table 1: The table compares our results with the best numerical estimates [63]$Z_\mu^{A,B}/v$ for Hamiltonian (43) and Hamiltonian (44).

| $J_z$ | $Z_\mu^A/v$ | our num. $Z_\mu^A/v$ | $Z_\mu^B/v$ | our num. $Z_\mu^B/v$ |
|---|---|---|---|---|
| 0.700 | 0.865 | 0.886 | 0.636 | 0.646 |
| 0.600 | 0.928 | 0.938 | 0.700 | 0.708 |
| 0.500 | 0.970 | 0.976 | 0.760 | 0.763 |
| 0.400 | 1.000 | 1.001 | 0.811 | 0.815 |
| 0.300 | 1.012 | 1.015 | 0.861 | 0.864 |
| 0.200 | 1.017 | 1.020 | 0.908 | 0.911 |
| 0.100 | 1.012 | 1.016 | 0.954 | 0.957 |
| 0.000 | 1.000 | 1.003 | 1.000 | 1.004 |
| -0.100 | 0.978 | 0.982 | 1.047 | 1.051 |
| -0.200 | 0.948 | 0.952 | 1.097 | 1.100 |
| -0.300 | 0.908 | 0.911 | 1.150 | 1.153 |
| -0.400 | 0.856 | 0.861 | 1.208 | 1.212 |
| -0.500 | 0.794 | 0.798 | 1.277 | 1.282 |

## 3.5 Determination of the renormalization factors

Instead of relying on the literature, we can of course determine $Z_\mu$ directly by studying the energy of the model. Indeed, using Hamiltonian (49) together with the result

$$\langle \cos \frac{\Phi}{R} \rangle = \frac{1}{(2\ell)^{2h}} \tag{68}$$

(recall $2h = g^{-1}$), leads immediately to the term $O(1)$ in energy

$$E = -\frac{\mu_R}{\pi} \frac{(-1)^\ell}{(2\ell)^{2h}} + O(\mu_R^2), \tag{69}$$

and thus to the difference between even and odd

$$\delta E = E^e - E^o = -2\frac{\mu_R}{\pi} \frac{1}{(2\ell)^{2h}} . \tag{70}$$

For a finite system we obtain the corresponding shift using a conformal mapping. If the total system is of length $L$ we have then

$$\langle \cos \frac{\Phi}{R}(\ell) \rangle = \left(\frac{\pi}{L}\right)^{2h} \frac{1}{[2 \sin \frac{\pi\ell}{L}]^{2h}} , \tag{71}$$

and thus for our case $Z = 2$ we find finally

$$\delta E = -2\frac{\mu_R}{\pi} \left(\frac{\pi}{4\ell}\right)^{2h} . \tag{72}$$

From the definition $\mu_R = Z_\mu \mu$ a numerical determination of $\delta E$ gives access to the renormalization factor (recall $Z_\mu = 1$ for $J_z = 0$).

Note that this time the sound velocity $v$ does not enter. The values of $Z_\mu$ for Hamiltonians (43) (44) determined this way are given below in table 2, and compared with those from [63], with excellent agreement.

Table 2: The table compares our determination of $Z_\mu^{A,B}$ obtained by fitting equation 72 with those form [63].

| $J_z$ | $Z_\mu^A$ | numerical $Z_\mu^A$ | $Z_\mu^B$ | numerical $Z_\mu^B$ |
|---|---|---|---|---|
| 0.700 | 1.220 | 1.222 | 0.892 | 0.890 |
| 0.600 | 1.258 | 1.256 | 0.948 | 0.946 |
| 0.500 | 1.259 | 1.258 | 0.985 | 0.984 |
| 0.400 | 1.237 | 1.237 | 1.007 | 1.007 |
| 0.300 | 1.198 | 1.198 | 1.018 | 1.019 |
| 0.200 | 1.143 | 1.144 | 1.020 | 1.021 |
| 0.100 | 1.076 | 1.078 | 1.014 | 1.015 |
| 0.000 | 1.000 | 1.002 | 1.000 | 1.002 |
| -0.100 | 0.915 | 0.917 | 0.980 | 0.982 |
| -0.200 | 0.823 | 0.825 | 0.952 | 0.954 |
| -0.300 | 0.726 | 0.727 | 0.918 | 0.921 |
| -0.400 | 0.622 | 0.624 | 0.877 | 0.880 |

## 4 Symmetries

We finally discuss in this section some symmetries of the problem, most of them occurring only in the scaling limit.

### 4.1 Symmetries between $\mu$ and $-\mu$, $\lambda$ and $-\lambda$

The entanglement entropy is expected to possess several interesting symmetries in the scaling limit. The first such symmetry can be seen from the point of view of the perturbed homogeneous chain, where we have seen in section 3.3 that in the field theory Hamiltonian (49), translation of the cut by one site amounts to $\mu_R \to -\mu_R$. Of course this is true only to first order in $\mu_R$, but since the results in the scaling limit are valid in the limit $\mu_R \to 0$, $\ell \to \infty$ with $\mu\ell^{1-g^{-1}}$ finite, it is only this order that matters. Hence we conclude that, in the scaling limit:

$$\delta S(\mu) = -\delta S(-\mu). \tag{73}$$

The second symmetry is simply

$$\delta S(\lambda) = \delta S(-\lambda). \tag{74}$$

This follows from the discussion of the perturbation expansion around the split fixed point, and the fact that to all orders $\delta S$ was found to be an even function of $\lambda$. The relationships (73) and (74) are illustrated in Fig. 14a and Fig. 14b respectively. Note that, as emphasized above, the relationships are only expected to hold in the scaling limit, $\mu \to 0$ (resp. $\lambda \to 0$) and $L \to \infty$ with the appropriate combinations $\Theta_B$ (resp. $T_B$) finite. As commented earlier, the spread of the curves in the IR is due to the difficulty of reaching the scaling limit while being technically limited to relatively small values of $L$.

### 4.2 Symmetries between $\lambda$ and $\frac{1}{\lambda}$

To see the third symmetry, imagine we consider a chain with $\lambda \gg 1$ i.e. with a coupling between sites $\ell$ and $\ell + 1$ greatly enhanced. To facilitate the discussion we introduce a slightly more general Hamiltonian for the interactions between sites $\ell - 1, \ell, \ell + 1$, and $\ell + 1, \ell + 2$:

$$\begin{aligned} H_\ell &= \sigma_{\ell-1}^x \sigma_\ell^x + \sigma_{\ell-1}^y \sigma_\ell^y + J_z \sigma_{\ell-1}^z \sigma_\ell^z + \lambda\left(\sigma_\ell^x \sigma_{\ell+1}^x + \sigma_\ell^y \sigma_{\ell+1}^y + \Delta \sigma_\ell^z \sigma_{\ell+1}^z\right) \\ &\quad + \sigma_{\ell+1}^x \sigma_{\ell+2}^x + \sigma_{\ell+1}^y \sigma_{\ell+2}^y + J_z \sigma_{\ell+1}^z \sigma_{\ell+2}^z, \end{aligned} \tag{75}$$

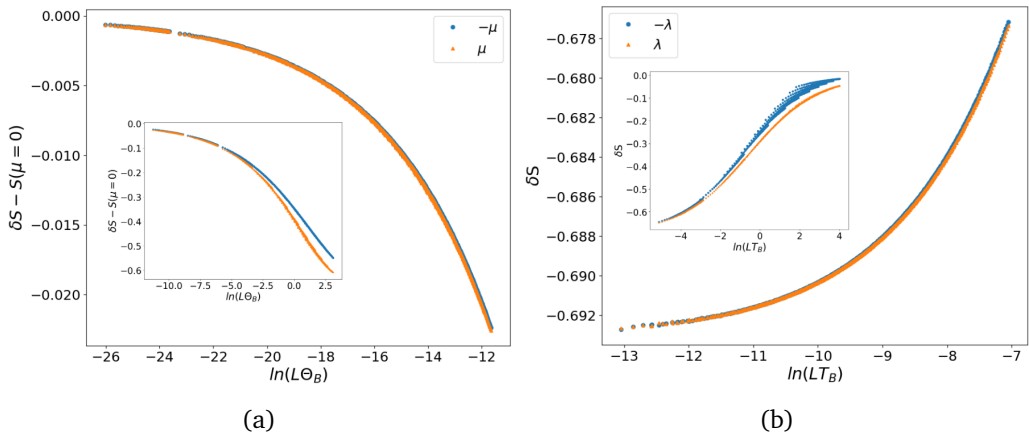

Figure 14: (a)Here $J_z=0.5$ and we compare $\delta S(\mu)$ and $-\delta S(-\mu)$ with Hamiltonian(43) when $\mu$ is very small and we are well in the scaling limit. (b) Here $J_z=-0.5$, and we compare $\delta S(\lambda)$ and $\delta S(-\lambda)$ with Hamiltonian(1) when $\lambda$ is very small. Insets are for a larger range of $L\Theta_B(LT_B)$.

where we have allowed for the coupling with amplitude $\lambda$ to have a different anisotropy $\Delta$ (instead of $J_z$). In the limit $\lambda \gg 1$, the spins $\vec{\sigma}_\ell$ and $\vec{\sigma}_{\ell+1}$ are almost paired into a singlet. The Hamiltonian can then be replaced in this limit, by its first-order perturbation theory approximation [70, 71]

$$H_\ell \mapsto -E_s + \sum_{t_i} \frac{|\langle s|H_\ell|t_i\rangle|^2}{E_s - E_{t_i}}, \tag{76}$$

where the energies of the term coupling spins $\ell$ and $\ell+1$ are $E_s, E_{t_i}$ respectively. For the singlet we have $E_s = -\lambda\left(\frac{1}{2} + \frac{\Delta}{4}\right)$ while the "triplet" now splits into states (for spins $\ell, \ell+1$) $|++\rangle$ and $|--\rangle$ with energies $E_{t_1} = E_{t_3} = \frac{\lambda\Delta}{4}$ and $\frac{|+-\rangle-|-+\rangle}{\sqrt{2}}$ with $E_{t_2} = \lambda\left(\frac{1}{2} - \frac{\Delta}{4}\right)$.

A straightforward calculation then gives, up to an irrelevant additional constant

$$H_\ell \mapsto \frac{1}{\lambda}\left(\frac{\sigma^+_{\ell-1}\sigma^-_{\ell+2} + \sigma^-_{\ell-1}\sigma^+_{\ell+2}}{1+\Delta} + \Delta^2\sigma^z_{\ell-1}\sigma^z_{\ell+2}\right). \tag{77}$$

Observe that, while initially the modified bond was between sites $\ell, \ell+1$, after this renormalization it is now between sites $\ell-1$ and $\ell+2$ which, after a relabelling starting as usual from the left, becomes between sites $\ell-1$ and $\ell$. Hence we have exchanged the odd and even impurity problems. Notice also that the anisotropy of the Hamiltonian is not in general preserved. This only occurs in the XXX case when $\Delta = 1$, for which we recover an XXX Hamiltonian, and the coupling has gone from $\lambda$ to $\frac{1}{2\lambda}$ and in the XX case when $\Delta = 0$ for which we recover an XX Hamiltonian but the coupling has gone from $\lambda$ to $\frac{1}{\lambda}$.

The duality is best seen for Hamiltonian $H_B$ (8) which corresponds to $\Delta = 0$. In this case we expect, in the scaling limit

$$\delta S(\lambda) = -\delta S\left(\frac{1}{\lambda}\right). \tag{78}$$

In general, since we have argued and checked that dependency of the $\delta S$ curve on the exact form of the modified Hamiltonian can entirely be absorbed into a redefinition of $T_B$, we expect the results for the problem and its dual to be identical (up to the exchange of odd and even) in the scaling limit. Moreover, in the case of Hamiltonians $H^A$ and $H^B$, the redefinition of $T_B$ can be obtained simply by the substitution $\lambda \to \frac{1}{\lambda(1+\Delta)}$. This relationship is illustrated in Fig. 15a, while the equation (78) is illustrated in Fig. 15b.

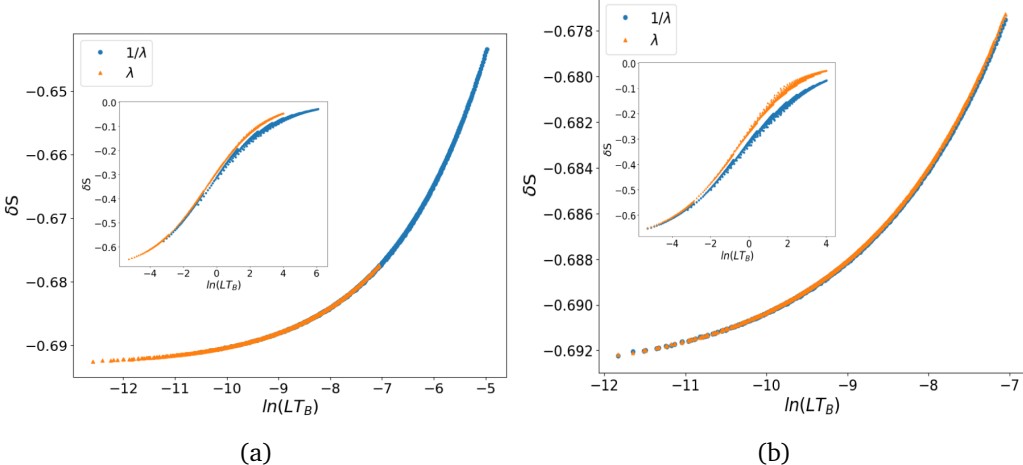

Figure 15: For these figures $J_z$=-0.5, and we compare $\delta S(\lambda)$ and -$\delta S\left(\frac{1}{\lambda}\right)$ when $\lambda$ is very small (and we are well in the scaling limit) (a) With Hamiltonian (1) and after the rescaling of the coupling by $\frac{1}{1+\Delta}$. (b) With Hamiltonian (8). In both cases the insert is for larger values of $LT_B$, where the scaling limit is not fully attained.

## 5 Conclusions

We have shown in this paper how odd-even effects in the Kane-Fisher problem persist even in the thermodynamic limit when considering terms $O(1)$ in the entanglement entropy of the XXZ chain, and how these effects encode a hybridization of the delocalized spin $1/2$ remaining in the odd case with the bath constituted by the rest of the chain. These results can as well be interpreted in the Luttinger liquid language in terms of an unpaired electron hybridizing with the electron bath, not unlike what happens in the Kondo problem. This gives rise to curves for $\delta S$ with qualitative features that are only dependent on the nature of the interaction (attractive or repulsive), and always interpolating between $-\ln 2$ and 0. Note however that details of these curves (such as the slopes at the origins etc) depend on the coupling constants.

The qualitative behavior of $\delta S$ is quite similar to e.g. the crossover occurring in the anisotropic Kondo problem when considering the dependency of the impurity entropy on the coupling and the temperature. We emphasize however that the results presented here are obtained at vanishing temperature, and explore rather the spatial dependency of the impurity screening. The extent to which finite size and finite temperature effects can be compared remains to be understood.

Although it may seem a mere technical point, we emphasize the remarkable fact that $\delta S$ is, at least to the order we have considered, accessible in UV perturbation theory. This is in sharp contrast with quantities such as the g-function [32]. The question of the full analytical structure of $\delta S$ as a function of $\ell T_B$ - and whether it can be calculated exactly using a technique such as the Bethe-Ansatz - remains open.

While our problem originated in the context of physics near a boundary, the parity effects we unveiled occur as well in the bulk. Consider indeed a periodic system of length $L$ and a sub-interval of length $\ell$ connected on both sides to the rest of the chain by modified bonds as in (1,8) - this is illustrated in Fig. 16 below. The physics (RG flow towards a healed or split chain) is expected to be the same as near a boundary. We find that the entanglement for subsystems of even or odd length (the figure corresponds to the latter case) also differs by terms of $O(1)$. The details of these terms are a bit intricate, and we plan to discuss them elsewhere. For now we contend ourselves with the following observation. In the non-interacting case $J_z = 0$ and for two slightly different couplings $\lambda$ and $r\lambda$ with $0 < r < 1$, the difference $\delta S$ at large $L$

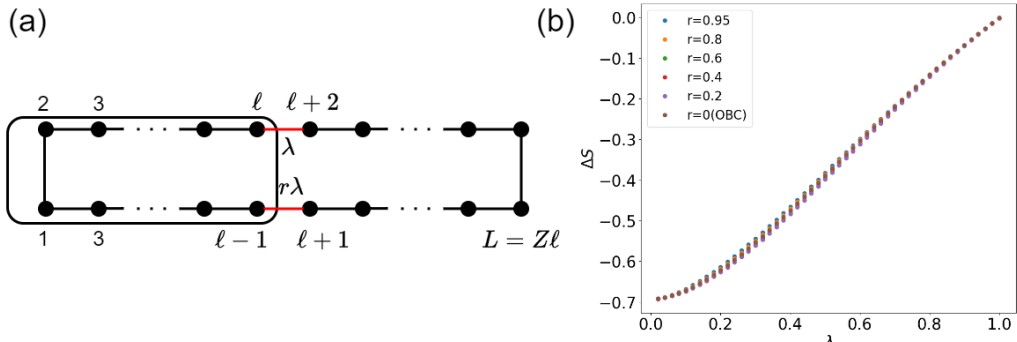

Figure 16: (a) Sketch of the PBC system. (b). Comparison of $\delta S$ for the systems with different values of the ratio $r$ (where the modified couplings on either side of the impurity are $\lambda$ and $r\lambda$. Of course, $r=0$ corresponds to a OBC system, $J_z=0$.

coincides, even in this periodic geometry, with the curve for the open geometry with a single modified bond $\lambda$: in other words, the weakest of the two modified bonds effectively behaves as if it were "opening" the system. While it is easy to understand this qualitatively (the system prefers to form valence bonds over the strongest bond), proving it analytically might be more difficult.

Another intriguing point is what the behavior of $\delta S$ can reveal about systems with extended arrays of modified bonds. In the non-interacting case, it was argued in [43] that the existence of a topological phase in the SSH model could be inferred from the behavior of $\delta S$. Whether a similar result holds in the presence of interactions remains to be seen, and will be discussed elsewhere.[12]

In conclusion, the fine structure of the entanglement in the presence of interactions reveals many interesting details, and remains to be explored more thoroughly. Going a bit out of the condensed-matter context, it is particularly intriguing to wonder what happens in the case of flows between topological defects [73], another topic we hope to discuss elsewhere.

## Acknowledgments

We thank H. Schloemer for related collaborations. HS thanks P. Calabrese and L. Capizzi for discussions. We would like to dedicate this work to the memory of I. Affleck, whose contribution to this topic (and so many others) was seminal.

**Funding information**   HS work was supported by the French Agence Nationale de la Recherche (ANR) under grant ANR-21- CE40-0003 (project CONFICA).

## A   Numerical calculation details

Our numerical results are obtained by using the TeNPy package [57], which is a Python library for the simulation of strongly correlated quantum systems with tensor networks. This package supports various algorithms like TEBD and DMRG. Specifically we used XXZChain model and two-site DMRG algorithm TwoSiteDMRGEngine. The maximum energy error is $10^{-10}$,

---

[12]We note in this respect that very few results are available for staggered XXZ chains (an exception is [72]).

the smallest Schmidt value is $10^{-10}$. In general the bond dimension is $\max(50, 0.5N)$, although in some cases such as $\lambda \approx 0$ and $J_z < -0.5$, the bond dimension is $\max(75, 0.75L)$, where $L$ is the system size. $L$ itself goes from 80 to 860. Finally, we use the package for the Spin-1/2 XXZ chain that imposes $S^z$ conservation. The initial state we used in all calculations was alternating spin up and spin down. We checked the final result did not depend on this choice.

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
