# Peer review of "Entanglement flow in the Kane-Fisher quantum impurity problem"

_SciPost Physics Core, doi:SciPost Phys. Core 8, 002 (2025)_

## Round 1 · Referee Report · Anonymous (Referee 1) · 2024-10-18

Report

The authors properly addressed all my minor comments plus all the others from the second referee. I think that now the paper is complete and ready for publication

Recommendation

Publish (easily meets expectations and criteria for this Journal; among top 50%)

---

## Round 1 · Referee Report · Anonymous (Referee 2) · 2024-11-4

Report

Let me start by acknowledging the effort by authors to address comments raised in my first report. The new version of the articled is a vast improvement. The article is now overall better structured, numerical details are present clearly, figures are more complete and captions are more descriptive, important concepts, terms and formula’s are formally introduced and/or properly described, the bibliography is extended, and numerous typos, grammatical errors and inconsistencies are resolved.

Nonetheless, the presentation can still be improved significantly in the introductions and discussion. Despite the authors having addressed most of its incompleteness in the introduction section, it still remains poorly structured and lacks motivation for why this problem is studied. The conclusion is in fact a summary. The authors quickly summarize the main findings and describe the effect of periodic boundary conditions, but an in-depth discussion of their findings and perspective is lacking. Furthermore, there are still errors, an example of which is the incorrect legend entries in figure 7, and numerous typos in the text.

Most important though, is whether this paper is suitable for SciPost Physics. My opinion that it is not. Observing Kane-Fisher flows in terms of O(1) in the entanglement is not a groundbreaking theoretical and computational discovery or a breakthrough on a previously-identified and long-standing research stumbling block. For sure the findings are interesting in itself, but they are a mere generalization of the parity effects observed in the entanglement entropy in XX chain studied by the authors and can therefore not really be considered a long-standing problem. Part of the tools used are the exact same with the addition of RG-flows one encounters in Kane-Fisher problems and the numerics are not of exceptional quality (the numerics clearly does not go beyond what is standard in the field).

Therefore, I recommend this article to be published in SciPost Core after considering the comments raised.

Recommendation

Accept in alternative Journal (see Report)

---

## Round 1 · Author Response

Dear Editor,

Please find enclosed a new version of our paper.

We thank both reviewers for their careful reading of the manuscript and their comments. Our paper received some criticisms by both referees concerning the presentation of the results, and also criticisms by the second referee concerning the substance and interest of our results. We would like to address these two aspects separately.

PRESENTATION

We are especially thankful to the second reviewer for demanding higher standards of pedagogy and clarity. We have striven in this new version to considerably improve the readability of the text following the advice of this reviewer.

We felt nevertheless that some of the demands were quite unreasonable, or betrayed, with all due respect, a distance from the field that might explain the severity of the criticisms. Examples of misunderstandings that could not in good faith be attributed to our lack of pedagogy are given below. As for “The manuscript is poorly structured with many errors and inconsistencies… “we are relieved to say we haven’t found a single “error” (short of rather innocuous misprints), and we still believe that the structure was carefully thought of, even if, indeed, we were rather cavalier with details. We hope this new version will be satisfactory.

We now go over these second reviewer points (concerning presentation) and address the corresponding changes in the new version of the manuscript (these changes are so numerous that it was impossible to provide a detailed list). The first reviewer points only concerned presentation, and were a (small) subset of the second referee's, so they are not addressed specifically, except for what concerns the size of the bibliography, which has grown to about 70 in this new version

The numbers below refer to the second reviewer's itemized list. This referees’ comments are written in bold for convenience.

1,2,3)

See section “Substance” below

**4) Numerical details are missing, making all numerical results non-reproduceable. This includes: Type of DMRG algorithm, bond dimension, convergence criteria, truncation error, number of sweeps, system sizes etc. **

Additional details about the numerics are now given in an appendix.

**5.) Lots of formulas are not formally introduced. The quantity of interest in this paper is the difference in the entanglement entropy between even and odd sectors. Besides the abstract it is nowhere introduced. Furthermore, this also holds for others important one. Take for example the definition of the constant appearing in the Affleck-Ludwig boundary entropy, shown in equation (2). This term includes boundary effects and is thus rather important for this study. Another example is the definition of the entropy in equation (34), which is not the well known formula for the Von-Neumann entropy and definitely requires the reference. **

We have striven to now have every formula (including the definition of what we call $\delta S$) properly introduced, sometimes with numbered equations.

As for the constant g in eq. (2), the referee seems mistaken. It is not the Affleck Ludwig (AL) boundary entropy but the Luttinger liquid parameter. AL entropy is not relevant for our purpose. A remark about this has been added before section 2.2. The discussion of the coupling constant around eq. (2) should now make this misunderstanding impossible.

We also do not fully understand the remark of the referee re the "non-standard definition" of the Von Neumann entropy. We use here the formulation of as the derivative of the Renyi entropies as a function of $n$ as $n \to 1$, which is totally standard. This is of course identical with the definition $S=-\left.\ln{1\over p-1}\ln R_p\right|_{p=1}$ due to the general definitions of derivatives of functions. . A footnote at the bottom of page 14 has been added to make this clear.

**6.) Details of the system is missing. An important part of the XXZ chain is the Luttinger Liquid that is realized for $|J_z|<1$. For someone that is known with either quantum spin chains or impurity problems it is obvious that the Luttinger Liquid is studied due to its free bosonic degrees of freedom and its corresponding logarithmic contributions to the entanglement entropy. But this is not clear for others. Furthermore, it puzzled me for a while whether the authors considered finite or semi-finite segment length. This should be clearly stated. **

We haven’t fully understood what the referee meant in this comment. The Luttinger liquid is not studied “due to its bosonic degrees of freedom” but because it is an archetype of many problems in condensed matter physics (now mentioned in the introduction with a large number of references). More details about the system are now given at the beginning of section 2.1.

**7.) Some parts of the findings are not elaborately touched on. Take for example equation 67. When $\Delta \to 1$the system approaches the phase boundary of the Luttinger Liquid with an exponential diverging Luttinger Liquid exponent K. This is reflected in this equation but is never described anywhere in the text. But this also hold for some behaviour in the numerical data. In figure 5 for example, there is a quite a bit of a discrepancy between the two curves. This has to be properly explained. **

Comments about the limit $\Delta \to -1$ are now given in the text. We are not particularly interested in this limit, where a marginal operator appears, together with logarithmic corrections, and numerics is known not to be particularly well behaved. We do not pretend to be exhaustive in this paper.

**8.) Important concepts and terms (though standard in the field) are not introduced, examples of which are healed and split, UV and IR, etc. **

All these concepts are now explained in detail, in particular at the end of the introduction, which also contains basic reminders about the renormalization group, scales etc

**9.) In section 3.3 the authors start to refer to equations in one of the references. It would simplify readers experience a lot if the authors provide these equations essential to understand part of the derivations. **

All the relevant equations from the reference are now reproduced - with relevant comments - in the present paper for the convenience of the reader.

**10) Caption are far to insufficient in describing the figures. The simply do not give any idea on what is presented. **

All captions have been considerably expanded.

**11) Sometimes the authors mention that results are checked numerically but not shown in the paper. Why are these not shown in an appendix? **

We decided not to show "all results" in an appendix to strike a balance between communicating new, and, we believe, interesting results, without necessarily being exhaustive. Interested readers can ask the authors for more detail on some fine points - we now mention this in the text. We have however, as a compromise, added figure 7 as requested by the referee.

12.) In the healed case the Hamiltonian $H^B$ contains an extra term $ J_{z} \sigma_{l}^{z} \sigma_{l+1}^{z}$ in comparison to the split case. There is no explanation, why it is so

The various choices of Hamiltonians are now discussed in the text, in particular before eq. (7) and before eq. (44).

**13.) Some results of Table 1 are not mentioned in the text. **

They should now all be mentioned.

**14.) Variables and constants appearing in formulas are quite often not described. **

We have striven to define properly all constants and quantities.

**15.) Quite a few typo’s in the text. Dots at the end of a sentence are missing. **

We have striven to correct all the misprints, though we humbly remain aware that perfection is not attainable in this world.

**16.) Inconsistencies and errors in the formulas such as capitals P in equation 32, |(0)+> instead of |(0)-> in equation 38, , same equation dash instead of double minus sign, etc. **

This has been corrected. We have worked to make this whole discussion clearer, and to correct the remaining misprints.

**17.) As mentioned by another referee, all equations appear in bold. **

This is now corrected

**18.) Missing DOI in quite a few references. **

This is now corrected.

SUBSTANCE

The second referee did not seem to doubt the interest of our paper, so we shall only reply to the first referee’s comments re this aspect.

**1,2,3) 1. Broader context of the results is missing. In large detail the results are derived and compared with numerical data. But there is no in-depth discussion, for instance, when numerical data demonstrate some deviation – what is the source of it, how to improve, etc? There is also no discussion on the broader impact of the results. 2. Introduction simply does not serve its role. No motivation is given, no context has been provided. When the authors write in the introduction “Think for instance of the Hamiltonian (1)” defined in the main body of the text it is extremely confusing. I understand that the authors worked on similar problems a lot and for them it is intuitive indeed to think about this Hamiltonian, but it is not as intuitive for the reader who have to see this Hamiltonian for the first time. In short, the manuscript requires reformatting and a better structured and clearer introduction to set the problem and a much more elaborate conclusion to place the findings into context, what they contribute to the problems mentioned above and proposals for further research. 3. What is the impact on specifically the Kane-Fisher problem never mentioned in the paper except in the introduction. **

Points 1,2 and 3 have been addressed in a much longer introduction (itself with subsections), rewritten abstract and added comments throughout the text.

We believe that observing the basic Kane -Fisher flows in terms of O(1) in the entanglement entropy is a new idea which suggests, in particular, the interesting interpretation of the "screening" of the left-over spin 1/2 in systems of odd length with the rest of the system, a phenomenon qualitatively similar to what happens in the Kondo problem but now, because the spin is delocalized over a length scale $\ell$, gives rise to non-trivial crossover curves. We consider it an important contribution to the field of entanglement and its behavior under RG flows, on the level of many other papers published in SciPost Physics on similar topics.

We also would like to stress that our paper makes the considerable effort of bridging high quality numerics with analytical results. Such results are extremely scarce in the literature, in particular because entanglement is often non-perturbative as discussed e.g. in reference [2] of the present manuscript. This means we had to strike a middle course in the discussion of details, and in devoting time to numerical issues which were not, in our opinion, particularly interesting (such as how to properly access the scaling limit when $\lambda$ is large and we are limited by accessible system sizes).

We hope that this new version of the paper will be acceptable in SciPost Physics as is, and thank once again the referees and the editor for their work on this.

---

## Round 1 · List of Changes

1. We have made overall improvements to the readability, we rewrote a much longer introduction, rewrote the abstract and added comments throughout the text, added explanations and definitions for important concepts and terms.
  2. we have added additional details about the numerics in an appendix, as well as throughout the text. We also have added a new figure about some numerical details.
  3. We have ensured that all formulas are properly introduced and defined.
  4. We have reproduced all equations necessary for the perturbative calculation of \delta S using perturbed conformal field theory, instead of merely referring to them by numbers in a related earlier paper of ours.
  5. We have considerably expanded all figure captions to better describe the figures.
  6. We have corrected all typos and misprints we were able to find
  7. We have considerably expanded the bibliography (going from 23 to 70 references)
  8. We have made sure the equations do not appear in bold any longer.
  9. We have added missing DOIs for all references (but one, which we were not able to find)

---

## Round 2 · Author Response

Dear Editor,

Please find enclosed a slightly updated version of the manuscript. We have not understood what the second referee thought was not correct in the caption of figure 7. To be on the safe side however, we have made this caption more explicit, and given additional detail on the meaning of the axes.

We also have added some simple comments in the conclusion to make it more substantial. We are sorry we are not really able to provide a "more in depth discussion and perspective” beyond this, but our study of terms of O(1) is one of very few, and it is not clear to us yet how deep it might be. We think it is better to be short and honest, rather than indulge in vague speculations.

For the sake of efficiency, we are happy to have our paper considered for Sci. Post Core instead of Sci. Post Phys. In view of us having at least one entirely positive referee report recommending publication in Sci. Post Phys., we hope this will lead to quick acceptance.

We thank you and the referees for your time and effort.

---

## Editorial Decision

published